# Dietary lipids fuel GPX4-restricted enteritis resembling Crohn's disease

Lisa Mayr[1,13], Felix Grabherr[1,13], Julian Schwärzler[1], Isabelle Reitmeier[1], Felix Sommer [2], Thomas Gehmacher[1], Lukas Niederreiter[1], Gui-Wei He[3], Barbara Ruder[3], Kai T. R. Kunz[1], Piotr Tymoszuk [4], Richard Hilbe[4,5], David Haschka[4], Clemens Feistritzer[6], Romana R. Gerner[1], Barbara Enrich[1], Nicole Przysiecki[1,7], Markus Seifert[4,5], Markus A. Keller [8], Georg Oberhuber[9], Susanne Sprung[10], Qitao Ran[11], Robert Koch[1], Maria Effenberger[1], Ivan Tancevski[4], Heinz Zoller[1], Alexander R. Moschen [1,7], Günter Weiss [4,5], Christoph Becker[3], Philip Rosenstiel[2], Arthur Kaser[12], Herbert Tilg[1] & Timon E. Adolph[1✉]

The increased incidence of inflammatory bowel disease (IBD) has become a global phenomenon that could be related to adoption of a Western life-style. Westernization of dietary habits is partly characterized by enrichment with the ω-6 polyunsaturated fatty acid (PUFA) arachidonic acid (AA), which entails risk for developing IBD. Glutathione peroxidase 4 (GPX4) protects against lipid peroxidation (LPO) and cell death termed ferroptosis. We report that small intestinal epithelial cells (IECs) in Crohn's disease (CD) exhibit impaired GPX4 activity and signs of LPO. PUFAs and specifically AA trigger a cytokine response of IECs which is restricted by GPX4. While GPX4 does not control AA metabolism, cytokine production is governed by similar mechanisms as ferroptosis. A PUFA-enriched Western diet triggers focal granuloma-like neutrophilic enteritis in mice that lack one allele of *Gpx4* in IECs. Our study identifies dietary PUFAs as a trigger of GPX4-restricted mucosal inflammation phenocopying aspects of human CD.

[1] Department of Internal Medicine I, Gastroenterology, Hepatology & Endocrinology, Medical University of Innsbruck, Innsbruck, Austria. [2] Institute of Clinical Molecular Biology, Christian-Albrechts-University Kiel and University Hospital Schleswig-Holstein, Kiel, Germany. [3] Department of Medicine 1, Gastroenterology, Pneumology and Endocrinology, University Medical Center Erlangen, Erlangen, Germany. [4] Department of Internal Medicine II, Infectious Diseases, Immunology, Rheumatology, Pneumology, Medical University of Innsbruck, Innsbruck, Austria. [5] Christian Doppler Laboratory for Iron Metabolism and Anemia Research, Medical University of Innsbruck, Innsbruck, Austria. [6] Department of Internal Medicine V, Haematology and Oncology, Medical University of Innsbruck, Innsbruck, Austria. [7] Christian Doppler Laboratory for Mucosal Immunology, Medical University of Innsbruck, Innsbruck, Austria. [8] Institute of Human Genetics, Medical University of Innsbruck, Innsbruck, Austria. [9] Pathology Department of Innsbruck Medical University Hospital, Innsbruck, Austria. [10] Department of Pathology, Medical University of Innsbruck, Innsbruck, Austria. [11] Department of Cell Systems and Anatomy, UT Health San Antonio, San Antonio, Texas, USA. [12] Division of Gastroenterology and Hepatology, Department of Medicine, Addenbrooke's Hospital, University of Cambridge, Cambridge, UK. [13] These authors contributed equally: Lisa Mayr and Felix Grabherr. ✉email: timon-erik.adolph@i-med.ac.at

Oxidation of biolipids, referred to as lipid peroxidation (LPO), is controlled by enzymatic (e.g. lipoxygenase-mediated) and non-enzymatic (e.g. fenton-type) reactions that particularly affect polyunsaturated fatty acids (PUFAs) within biological membranes[1–3]. LPO impairs cellular functions partly by forming cytotoxic protein adducts or damage to cellular membranes[4]. Compensation is provided by glutathione peroxidase 4 (GPX4), the only selenoprotein that catalyzes the reduction of oxidized biolipids[1,3,5]. Deletion of both Gpx4 alleles in mice or pharmacologic GPX4 inhibition in cells induces a distinct regulated form of iron-dependent cell death termed ferroptosis[1]. Ferroptosis requires acyl-CoA synthetase long-chain family member 4 (ACSL4)-mediated membrane enrichment of the ω-6 PUFA arachidonic acid (AA), which is prone to oxidation[2,6,7]. Deletion of both alleles of Gpx4 culminates in organ injury of the kidney, brain, and skin, which is conceivably elicited or modulated by immune responses[3,8–11]. While studies have identified key regulators of GPX4-restricted LPO and cellular demise[1,2,6,12–14], mechanism(s) of concurrent inflammatory responses remain elusive.

Inflammatory bowel diseases (IBDs) and specifically Crohn's disease (CD) are characterized by chronic remittent intestinal inflammation that arises from complex interactions between environmental factors (e.g. diet) in a genetically susceptible host[15]. However, plausible examples to support this assumption remain scarce[16–18]. Notably, the increase in incidence of IBD parallels the increase in dietary intake of ω-6 PUFAs such as AA, which is a major component of a Western diet and contained in meat and eggs[19]. Although AA intake entails a risk for developing IBD[20] and accumulates in the inflamed mucosa of IBD patients[21], the impact of AA and PUFA metabolism on intestinal inflammation remains controversial[22].

Given the genetic association between GPX4 and CD[23] and reports of GPX4-restricted AA oxidation in biological membranes[2,6], we set out to study the role of intestinal epithelial GPX4 in controlling gut homeostasis[24,25]. We find that CD epithelium exhibits reduced GPX4 activity and features of LPO. In intestinal epithelial cells (IECs) with reduced GPX4 activity, PUFAs and specifically AA induce the release of interleukin 6 (IL-6) and chemokine (C-X-C motif) ligand 1 (CXCL1) which is governed by iron availability, lipoxygenase-mediated LPO and Acsl4. Mice that are exposed to a PUFA-enriched Western diet and that lack one Gpx4 allele in IECs (Gpx4$^{+/−IEC}$), but not wild-type (WT) mice, display signs of epithelial LPO and focal neutrophilic enteritis with granuloma-like accumulation of inflammatory cells. Oral AA exposure evokes neutrophilic inflammation in the small intestine of iron-primed Gpx4$^{+/−IEC}$ mice. Enteritis in both models can be ameliorated by LPO scavenging. As such, our study exemplifies how PUFAs in a Western diet pose a risk for developing CD.

## Results

### Impaired epithelial GPX4 activity features CD. 
To investigate a role of reduced GPX4 activity and LPO in human IBD, we analyzed biopsy-derived IEC-enriched specimens from the lesional and non-lesional mucosa of CD and ulcerative colitis (UC) patients with active disease. Non-IBD patients who underwent screening colonoscopy and lacked demonstrable intestinal disease by endoscopic and histologic means served as healthy controls (HC). Clinical characteristics of this cohort are summarized in Table 1. IECs derived from the lesional small intestinal mucosa of CD patients exhibited decreased expression of GPX4, which was paralleled by decreased enzymatic activity (Fig. 1a-e and Supplementary Fig. 1A). In contrast, colonic GPX4 expression and activity in UC patients was indistinguishable from that in healthy controls (Fig. 1f, g), similar to GPX4 expression in colonic CD (Fig. 1f). In line with this, IECs of the lesional small intestinal mucosa of CD patients exhibited signs of LPO indicated by 4-HNE adducts (Fig. 1h), which was similarly notable in small intestinal epithelial organoids retrieved from lesional mucosa of CD patients (Supplementary Fig. 1B).

### PUFAs evoke an inflammatory response of Gpx4-deficient IECs. 
To analyze the role of GPX4 in IECs, we first generated Gpx4$^{−/−}$ small intestinal epithelial MODE-K cells by CRISPR Cas9 editing of exon 1[26], which induced IEC death and thus prevented further studies (Supplementary Fig. 1C, D). To assess the consequences of reduced (but not completely abrogated) GPX4 activity in IECs, we silenced MODE-K cells with Gpx4 small-interfering RNA (siGpx4). siGpx4 silencing impaired Gpx4 expression and enzymatic activity by ~75% (Supplementary Fig. 1E-G). As oxidation of PUFAs and specifically AA is restricted by GPX4[2], we, in a first step, tested the impact of reduced GPX4 activity and AA exposure on intestinal epithelial LPO. Indeed, the ω-6 PUFA AA deteriorated LPO in Gpx4-deficient IECs (Fig. 2a and Supplementary Fig. 1H). Importantly, AA induced the expression of IL-6 and CXCL1 in siGpx4 IECs, but not in control IECs (Fig. 2b–e). Similarly, ω-3 and ω-6 PUFAs, i.e. stearidonic acid (SDA), docosahexaenoic acid (DHA), eicosapentaenoic acid (EPA) and docosapentaenoic acid (DPA), induced LPO (Fig. 2f) and IL-6 and CXCL1 production (Fig. 2g, h) in siGpx4, but not in siCtrl IECs. The saturated long-chain fatty acid palmitic acid (PA) induced LPO, IL-6 and CXCL1 responses to a similar extent in siGpx4 and siCtrl IECs (Fig. 2f-h). Monounsaturated fatty acids such as palmitoleic acid (POA) and oleic acid (OA) did not impact LPO or cytokine production in siGpx4 IECs (Fig. 2f-h). Other pro-inflammatory stimuli such as TNFα or IL-1β evoked cytokine responses from siGpx4 IECs that were comparable to siCtrl IECs (Fig. 2i, j). As such, PUFAs particularly elicited LPO and a cytokine response in Gpx4-deficient IECs.

### A PUFA-enriched Western diet induces enteritis in Gpx4$^{+/−IEC}$ mice. 
Next, we set out to study the impact of PUFAs on intestinal inflammation in GPX4-deficient mice. We were unable to retrieve homozygous Gpx4$^{flox/flox}$;Villin-Cre$^{+/−}$ (Gpx4$^{−/−IEC}$) mice as the offspring died in utero. However, 11% Gpx4$^{−/−IEC}$ pups were born when the diet of the mothers during gestation

| Table 1 Patient characteristics. | | | |
|---|---|---|---|
| | HC | CD | UC |
| Total N | 21 | 16 | 12 |
| male/female (%) | 43/57 | 50/50 | 75/25 |
| Age (years) | 51.62 ± 12.1 | 39.06 ± 14.4 | 40.38 ± 13.7 |
| Body mass index (kg/m²) | 28.56 ± 6.17 | 24.06 ± 5.81 | 23.70 ± 3.48 |
| MAYO Score | N/A | N/A | 4.75 ± 0.69 |
| Harvey–Bradshaw Index | N/A | 4,67 ± 0.52 | N/A |
| CRP (mg/dl) | N/A | 0.55 ± 0.04 | 0.59 ± 0.14 |

HC healthy control, CD Crohn's disease, UC ulcerative colitis, N patient numbers, CRP C-reactive protein.

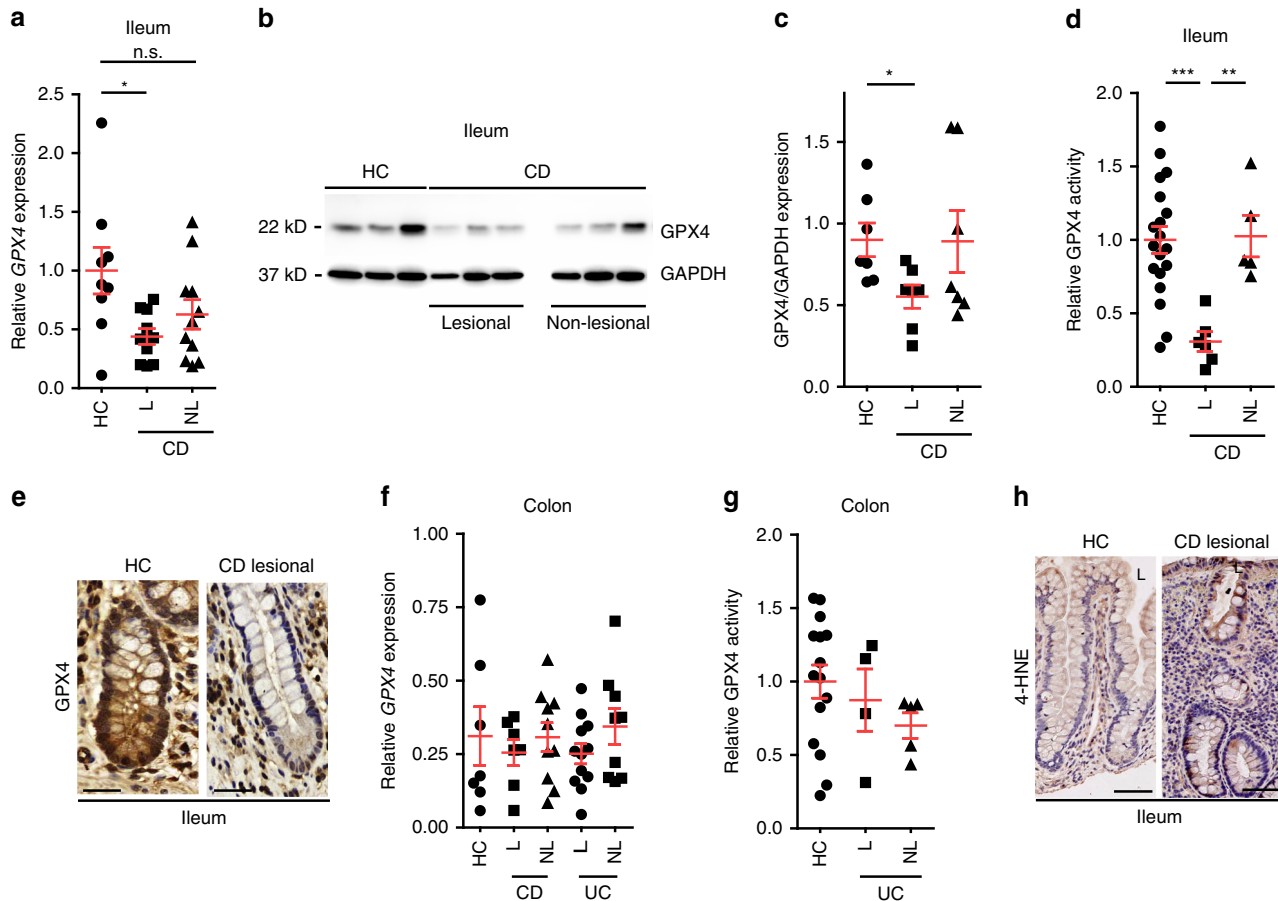

**Fig. 1 Reduced GPX4 activity and LPO localize to IECs in CD patients. a** Relative *GPX4* expression in the macroscopically inflamed (lesional, L) and macroscopically non-inflamed (non-lesional, NL) small intestinal mucosa of CD patients determined by qPCR and compared to healthy controls (HC). Each dot represents one patient ($n = 9$ for HC, $n = 11$ for CD-L, and $n = 10$ for CD-NL). *$P = 0.0167$. **b, c** Representative GPX4 immunoblot of IEC-enriched fractions from biopsies taken from the ileum of CD patients and HC (**b**), with densitometry relative to GAPDH shown in (**c**). Each dot represents one patient ($n = 7$ patients per group). *$P = 0.0265$. **d** Relative GPX4 enzymatic activity of epithelial-enriched fractions derived from the lesional (L) and non-lesional (NL) mucosa of the small intestine of CD patients as compared to HC. Each dot represents one patient ($n = 19$ for HC, $n = 6$ for CD-L, and $n = 5$ for CD-NL). ***$P < 0.001$, **$P < 0.01$. **e** Representative GPX4 immunoreactivity (brown) determined in the lesional small intestinal sections of CD patients as compared to HC ($n = 7$). **f** Relative *GPX4* expression in the macroscopically inflamed (lesional, L) and non-inflamed (non-lesional, NL) mucosa of the large intestine of CD patients and UC patients as compared to HC. Each dot represents one patient ($n = 7$ for HC, $n = 9$ CD-L, $n = 10$ for CD-NL, $n = 12$ for UC-L and $n = 7$ for UC-NL). **g** Quantification of GPX4 enzymatic activity in lesional and non-lesional IEC-enriched fractions from the large intestine of UC as compared to HC. Each dot represents one patient ($n = 15$ for HC, $n = 4$ for UC-L, and $n = 5$ for UC-NL). **h** 4-HNE immunoreactivity (brown) in HC and small intestinal lesional sections of CD patients, indicative for LPO. L, luminal-oriented side ($n = 3$ patients per group). Scale bars indicate 25 μm (**e**) and 50 μm (**h**). For panel (**a**), (**c**), (**d**), (**f**), and (**g**) data are presented as mean±SEM. One-way ANOVA with Bonferroni's multiple comparison test. Source data are provided as a Source Data file.

was supplemented with α-tocopherol (Supplementary Table 1, Supplementary Fig. 2A, B). *Gpx4*$^{-/-IEC}$ pups from α-tocopherol treated mothers had lower weight at birth but regained weight at 7 weeks of age (Supplementary Fig. 2C–F), with a mucosa that was morphologically comparable to WT littermates (Supplementary Fig. 2G, H). For further studies, we utilised *Gpx4*$^{flox/wt}$; *Villin-Cre*$^{+/-}$ (*Gpx4*$^{+/-IEC}$) mice[27,28], which specifically deleted one *Gpx4* allele in the intestinal epithelium and resulted in a ~50% reduction of mRNA and protein levels in small and large intestinal IECs (Supplementary Fig. 3A, B). *Gpx4*$^{+/-IEC}$ mice were viable, born at Mendelian ratio and exhibited a mucosal appearance indistinguishable from that of WT littermates (Fig. 3a and Supplementary Fig. 3C). LPO, IEC death and proliferation were comparable between *Gpx4*$^{+/-IEC}$ and WT mice (Supplementary Fig. 3D–H). Notably, *Gpx4*$^{+/-IEC}$ mice were susceptible to colonic inflammation induced by dextran sodium

sulfate (DSS) (Supplementary Fig. 3I–M), as previously observed in mice with myeloid-specific deletion of *Gpx4*[29].

In a next step, we orally challenged *Gpx4*$^{+/-IEC}$ and WT mice with a Western-style diet (ssniff TD88137) enriched with or without 10% fish oil (containing ω-3 and ω-6 PUFAs, Supplementary Table 2) for 3 months. As expected, WT mice were unaffected by a low-fat control diet (LFD), a Western diet (WD) or a PUFA-enriched Western diet (PUFA WD) (Fig. 3b and Supplementary Fig. 4A). In contrast, a PUFA-enriched Western diet evoked patchy small intestinal inflammation in *Gpx4*$^{+/-IEC}$ mice, while the colon was unaffected. Specifically, PUFA WD-fed *Gpx4*$^{+/-IEC}$ mice displayed mucosal to submucosal infiltration of neutrophil granulocytes and mononuclear cells, crypt hyperplasia, epithelial injury and granuloma-like accumulation of inflammatory cells resembling some aspects of small intestinal CD (Fig. 3b–d and Supplementary Fig. 4B). Intestinal inflammation in PUFA WD-fed *Gpx4*$^{+/-IEC}$

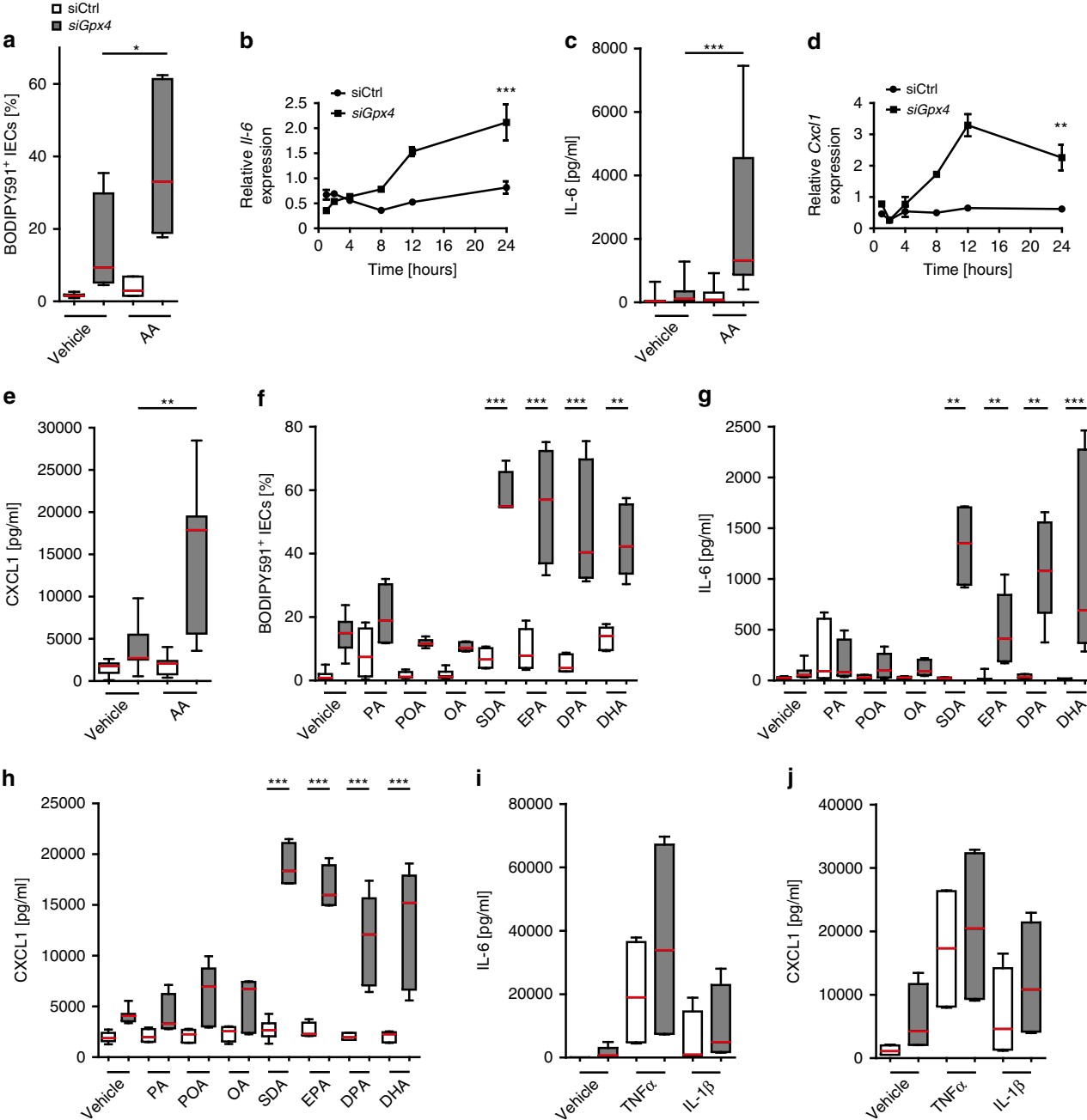

**Fig. 2 PUFAs trigger epithelial LPO and an inflammatory response restricted by GPX4. a** LPO quantification by flow cytometry of BODIPY581/591 C11+-labeled IECs stimulated with arachidonic acid (AA) for 24 h (*n* = 6 biologically independent experiments). *P = 0.0183. **b, c** Quantification of IL-6 expression from *siGpx4* and siCtrl IECs over a course of AA stimulation determined by qPCR (*n* = 4 biologically independent experiments), ***P<0.001 (**b**), and after 24 h by ELISA (*n* = 12 for vehicle and *n* = 24 for AA biologically independent experiments). ***P < 0.001 (**c**). **d, e** Quantification of CXCL1 expression from *siGpx4* and siCtrl IECs over a course of AA stimulation determined by qPCR (*n* = 3 biologically independent experiments). **P = 0.001 (**d**) and after 24 h by ELISA (*n* = 12 for vehicle and *n* = 24 for AA), **P = 0.0031 (**e**). **f** LPO quantification by flow cytometry of BODIPY581/591 C11+-labeled IECs stimulated with the saturated fatty acid palmitic acid (PA), monounsaturated fatty acids palmitoleic acid (POA) and oleic acid (OA) and polyunsaturated fatty acids stearidonic acid (SDA), eicosapentaenoic acid (EPA), docosapentaenoic acid (DPA) and docosahexaenoic acid (DHA) for 24 h (*n* = 4–12 biologically independent experiments) For all: ***P < 0.001 and **P<0.01 (**g, h**). Quantification of IL-6 and CXCL1 expression from *siGpx4* and siCtrl IECs stimulated with PA, POA, OA, SDA, EPA, DPA, and DHA for 24 h by ELISA (*n* = 4–11 biologically independent experiments). ***P < 0.001 and **P < 0.01. **i, j** Quantification of IL-6 and CXCL1 production from *siGpx4* and siCtrl IECs stimulated with TNFα or IL-1β for 24 h by ELISA (*n* = 4 biologically independent experiments). For panel (**a**), (**c**) and (**e**–**j**) data are presented as boxblot with median and interquartile range (25th and 75th). The whiskers represent minimal and maximal values. For panel (**b**) and (**d**) data presented as mean ± SEM. For panel (**a**–**j**) to (**j**) one-way ANOVA with Bonferroni multiple comparison test or a Kruskal–Wallis test with Dunn's multiple comparison test was used with the exception of panel (**b**) and (**d**) for which a two-way ANOVA with Bonferroni post-hoc test was used and panel (**f**) and (**g**) for which an unpaired two-tailed Student's *t* test was used. Source data are provided as a Source Data file.

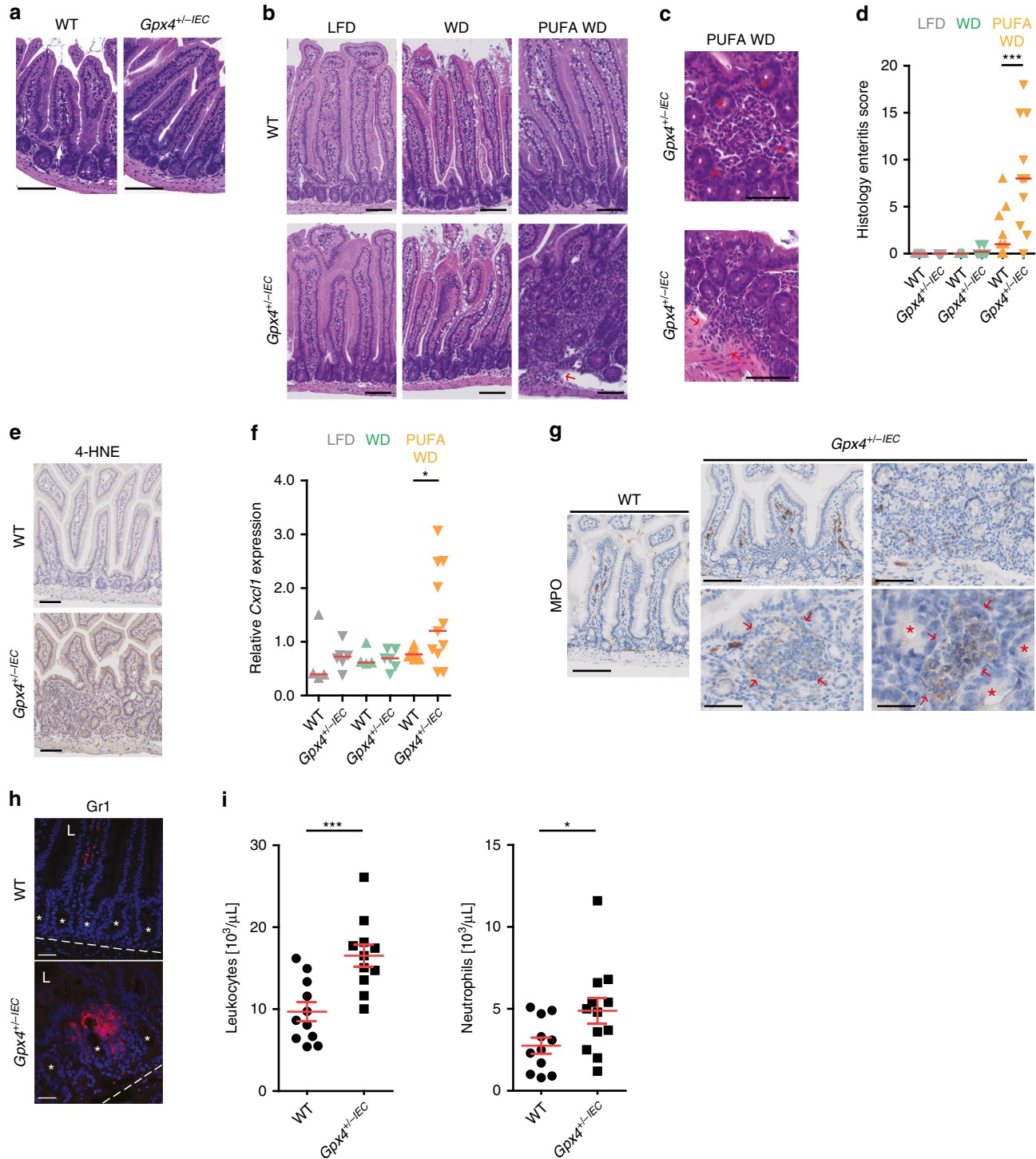

mice was characterized by signs of epithelial LPO (Fig. 3e), expression of *Cxcl1* (Fig. 3f) and infiltration of MPO[+] and GR1[+] neutrophils (Fig. 3g, h). *Gpx4*[+/−IEC] mice also exhibited higher levels of circulating leukocytes and specifically neutrophil granulocytes in the blood (Fig. 3i). We observed a similar enteritis severity in male and female *Gpx4*[+/−IEC] mice (Supplementary Fig. 4C). Notably, a Western diet (rich in saturated fatty acids) did not induce intestinal inflammation in *Gpx4*[+/−IEC] mice (Fig. 3b, d and Supplementary Fig. 4D). Although PUFA stimulation (but not cytokines, bile acids or lipopolysaccharide) impaired epithelial GPX4 activity likely because of proteasomal degradation of GPX4 (Supplementary Fig. 5A-H) and *Gpx4* deficiency induced ferroptosis of MODE-K

IECs to some extent (Supplementary Fig. 6A-I), we did not observe IEC death in *Gpx4*[+/−IEC] mice on a PUFA-enriched Western diet (Supplementary Fig. 6J-L). These data indicated that cell death was not a prerequisite of PUFA-induced and GPX4-restricted intestinal inflammation in *Gpx4*[+/−IEC] mice.

**Cytokine production is driven by ferroptosis mechanisms.** We focused on regulatory mechanisms that control AA-induced cytokine production because AA entails a risk for developing IBD[20,21]. As we noted that AA stimulation promoted LPO only in *Gpx4*-deficient IECs (Fig. 2a) and cellular iron availability and lipoxygenases (LOX) control LPO in *Gpx4*-deficient cells[1,30], we

**Fig. 3 PUFA enrichment of a Western diet induces focal enteritis in *Gpx4*$^{+/-IEC}$ mice. a** Representative small intestinal H&E images of *Gpx4*$^{+/-IEC}$ mice and WT littermates on a chow diet ($n = 10$ mice per group). Scale bars indicate 100 µm. **b** Representative H&E images of WT and *Gpx4*$^{+/-IEC}$ mice exposed to a low-fat diet (LFD), a Western diet (WD) or a PUFA-enriched WD (PUFA WD) for 3 months. Note that the PUFA WD evoked focal enteritis characterized by mono- and polymorphonuclear cell infiltration, crypt hyperplasia, and mucosal injury in *Gpx4*$^{+/-IEC}$ mice ($n = 8$ mice for WT LFD, $n = 9$ mice for *Gpx4*$^{+/-IEC}$ LFD, $n = 7$ mice for WT WD, $n = 10$ mice for *Gpx4*$^{+/-IEC}$ WD, $n = 9$ mice for WT PUFA WD and $n = 11$ mice for *Gpx4*$^{+/-IEC}$ PUFA WD). Scale bars indicate 100 µm. **c** Granuloma-like accumulation of inflammatory cells and submucosal infiltration of inflammatory cells in *Gpx4*$^{+/-IEC}$ mice exposed to a PUFA-enriched WD for 3 months (red arrows) ($n = 11$). Scale bars indicate 100 µm. **d** Histology score of WT and *Gpx4*$^{+/-IEC}$ mice exposed to a low-fat diet (LFD), a Western diet (WD) or a PUFA-enriched WD (PUFA WD) for 3 months. Each dot indicates one experimental animal ($n = 8$ mice for WT LFD, $n = 9$ mice for *Gpx4*$^{+/-IEC}$ LFD, $n = 7$ mice for WT WD, $n = 10$ mice for *Gpx4*$^{+/-IEC}$ WD, $n = 9$ mice for WT PUFA WD and $n = 11$ mice for *Gpx4*$^{+/-IEC}$ PUFA WD). Median shown, \*\*\*$P < 0.001$. One-way ANOVA with Bonferroni's multiple comparison test. **e** Representative 4-HNE immunoreactivity (brown), indicative for LPO ($n = 5$ mice per group). Scale bars indicate 100 µm. **f** Relative *Cxcl1* expression determined by qPCR ($n = 4$ mice for WT LFD, $n = 6$ mice for *Gpx4*$^{+/-IEC}$ LFD, $n = 4$ mice for WT WD, $n = 6$ mice for *Gpx4*$^{+/-IEC}$ WD, $n = 8$ mice for WT PUFAWD, and $n = 11$ mice for *Gpx4*$^{+/-IEC}$ PUFA WD). \*$P = 0.0288$. One-way ANOVA with Bonferroni correction. **g** Representative images of MPO$^+$ cells in *Gpx4*$^{+/-IEC}$ and WT mice on a PUFA-enriched WD ($n = 5$ mice per group). The red arrows denote granuloma-like lesions with MPO$^+$ cells (brown) intermingled between crypts marked with asterisks. Scale bars indicate 50 µm and 100 µm, respectively. **h** Representative confocal images of GR-1$^+$ neutrophils (red) in *Gpx4*$^{+/-IEC}$ and WT mice on a PUFA-enriched WD ($n = 4$ mice per group). DAPI stained nuclei blue. Dashed line denotes basal membrane; L, luminal-oriented side. Asterisks denote crypt units. Scale bars indicate 50 µm. **i** Leukocyte count and neutrophil granulocyte count from whole blood samples of *Gpx4*$^{+/-IEC}$ and WT mice after a 3-month PUFA-enriched WD (Left panel $n = 11$ mice, right panel $n = 11$ mice for WT and $n = 12$ mice for *Gpx4*$^{+/-IEC}$). \*$P = 0.0361$, \*\*\*$P < 0.001$. Unpaired two-tailed Student's *t* test. Source data are provided as a Source Data file.

analyzed the impact of iron and LPO on GPX4-restricted cytokine production. Ferric iron promoted LPO (Fig. 4a) and AA-induced cytokine production in *siGpx4* IECs (Fig. 4b, c), despite a reduced iron uptake that could not be explained by differential regulation of iron transporters (Supplementary Fig. 7A–G). Vice versa, iron chelation with deferoxamine (DFO) reduced IL-6 and CXCL1 responses of *siGpx4* IECs upon AA stimulation (Fig. 4d, e). Moreover, AA-induced cytokine responses were ameliorated by the LPO scavenger ferrostatin-1 and α-tocopherol (Fig. 4f–h). Similarly, inhibition of LOX15 ameliorated AA-induced LPO (Supplementary Fig. 7H) and cytokine production of *siGpx4* IECs (Fig. 4i, j). Co-silencing of *Alox15* (or *Alox12*) also ameliorated the cytokine response of *siGpx4* IECs after AA stimulation (Fig. 4k and Supplementary Fig. 7I, J). The non-selective COX1/2 inhibitor piroxicam abolished neither CXCL1 production nor LPO (Supplementary Fig. 7K, L). Collectively, these data demonstrate that iron availability promoted PUFA-induced LPO and cytokine production in IECs with reduced GPX4 activity, which was reversed by α-tocopherol.

**AA and ferric maltol induce enteritis in *Gpx4*$^{+/-IEC}$ mice.** These findings led us to study the direct impact of AA and ferric iron on intestinal inflammation. Indeed, intestinal epithelial organoids from *Gpx4*$^{+/-IEC}$ mice, but not WT controls, displayed signs of LPO and increased *Cxcl1* expression on AA and ferric iron exposure (Fig. 5a, b and Supplementary Fig. 8a), while unstimulated *Gpx4*$^{+/-IEC}$ organoids were indistinguishable from WT organoids (Supplementary Fig. 8B-G). Next, we orally challenged 8-week old WT and *Gpx4*$^{+/-IEC}$ mice with AA once daily for five consecutive days complementary to the standard chow diet in addition to iron supplementation with ferric maltol (Fig. 5c), which is approved for treatment of iron-deficiency anemia in IBD[31]. We used this model as it may reflect a daily PUFA and iron challenge in humans on a meat-enriched Western diet[19]. WT mice were unaffected by oral AA and ferric maltol exposure (Fig. 5d, e). In contrast, *Gpx4*$^{+/-IEC}$ mice exhibited inflammation in the intestine when exposed to AA and ferric maltol (Fig. 5d, e). More specifically, acute inflammation in *Gpx4*$^{+/-IEC}$ mice was characterized by neutrophil infiltration in the proximal small intestine (Fig. 5d, e), the locale of PUFA and iron absorption[32,33], as corroborated by flow cytometry of GR1$^+$ neutrophils (Fig. 5f) and immuno-labeling of MPO$^+$ cells (Supplementary Fig. 8H). Neutrophilic infiltration was paralleled by signs of epithelial LPO (Fig. 5g) and increased

*Cxcl1* expression (Fig. 5h). Notably, ferric maltol or AA exposure alone did not evoke neutrophilic inflammation in *Gpx4*$^{+/-IEC}$ mice (Fig. 5e). The abundance of other mucosal innate and adaptive immune cells remained comparable between WT and *Gpx4*$^{+/-IEC}$ mice exposed to AA and ferric maltol (Supplementary Fig. 9A–H) and we did not note colonic inflammation (Supplementary Fig. 10A). These data demonstrated that intestinal epithelial GPX4 restrained neutrophilic small intestinal inflammation induced by AA and ferric maltol. In line with a critical role of LPO in inflammation, α-tocopherol treatment protected against PUFA-induced LPO and cytokine production (Fig. 4f–h) and neutrophilic infiltration in *Gpx4*$^{+/-IEC}$ mice challenged with AA and ferric maltol (Fig. 5i). Similarly, α-tocopherol, as well as liproxstatin-1 treatment, protected against enteritis in *Gpx4*$^{+/-IEC}$ mice induced by a PUFA WD (Fig. 5j, k), which was associated with reduced signs of LPO and neutrophil infiltration (Supplementary Fig. 10B–D).

**Cytokine production is governed by ACSL4.** To further explore how AA may instigate cytokine production, we used liquid chromatography tandem mass spectrometry (LC-MS/MS) to investigate the AA metabolite profile of *siGpx4* IECs as compared to that of controls. AA may be metabolized by cyclooxygenases (COX), lipoxygenases (LOX) and cytochrome P450 enzymes to bioactive lipid mediators, which occurred to a similar extent in AA-stimulated *siGpx4* and siCtrl IECs (Fig. 6a–c, Supplementary Table 3). In line with this, *Lox* and *Cox* expression was comparable between siCtrl and *siGpx4* IECs (Supplementary Fig. 11A). As *Acsl4* is required for IEC ferroptosis (Supplementary Fig. 6I)[2,6], we next hypothesized that AA-induced inflammation in *siGpx4* IECs required ACSL4. Indeed, *Acsl4* deletion abolished AA-induced IL-6 and CXCL1 production in *siGpx4* IECs (Fig. 6d, e).

Next, we sought to understand how ACSL4 controlled AA-induced cytokine responses. Notably, *Acsl4* deletion in *siGpx4* IECs did not protect against LPO after AA exposure (Supplementary Fig. 11B). These data indicated that ACSL4 controlled AA-induced cytokine production using a distinct mechanism that was independent of LPO. To explore a role of ACSL4 in AA metabolism, we analyzed the metabolite profile in *Acsl4*$^{-/-}$ IECs by means of LC-MS/MS (Fig. 6a-c). Indeed, *Acsl4*$^{-/-}$ IECs exhibited a decreased abundance of LOX and COX metabolites after AA stimulation in siCtrl and *siGpx4* IECs (Fig. 6a, b).

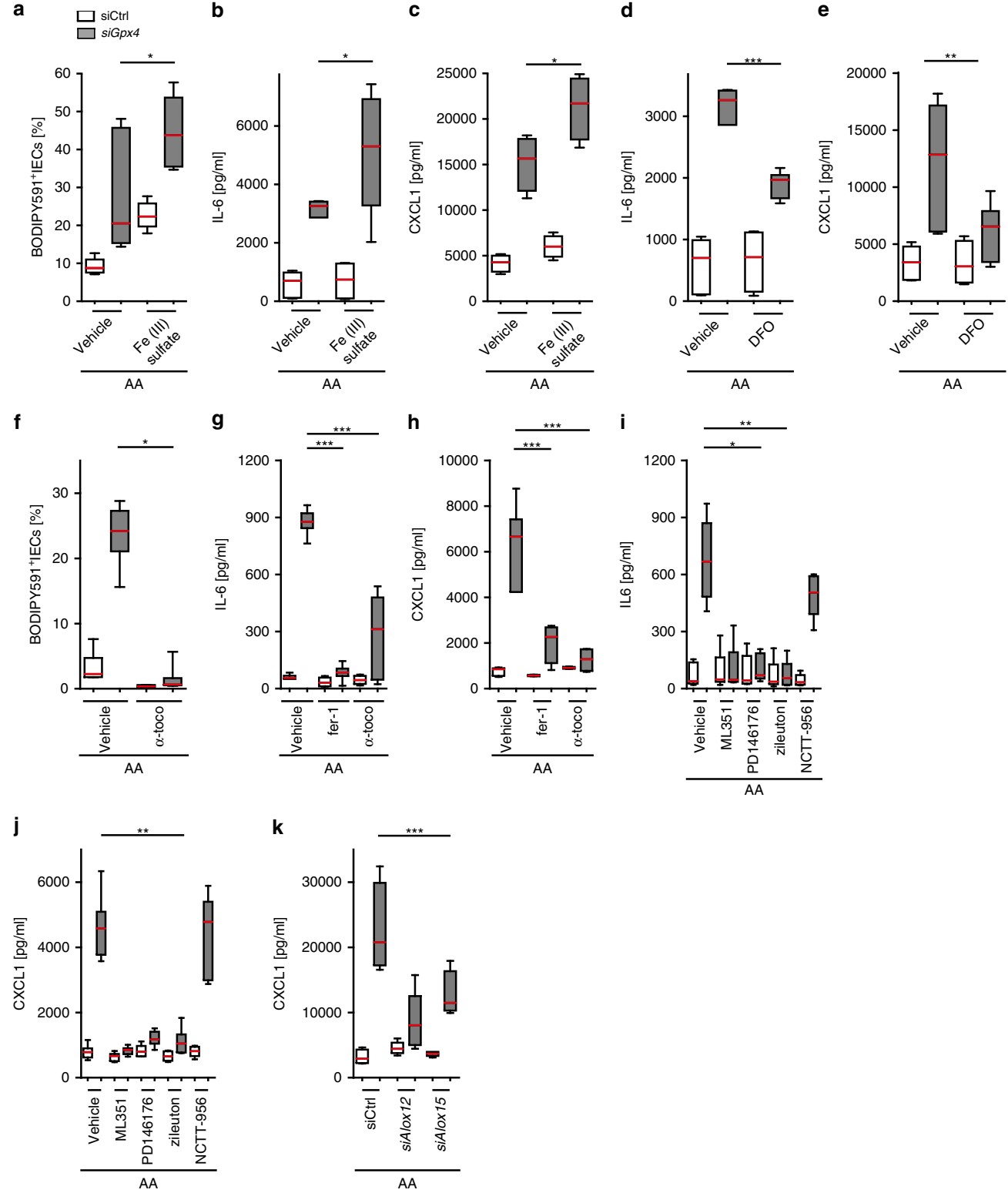

However, none of the abundant LOX and COX lipid mediators (i.e. 5-HETE, a 15-HETE precursor or PGE2) was able to promote the production of IL-6 or CXCL1 in our model (Supplementary Fig. 11C–H). These data suggested that modulation of LOX and COX metabolism by ACSL4 did not affect the inflammatory tone. However, we noted that P450 metabolites of AA (i.e. epoxyeicosatrienoic acids or 'EETs') were increasingly accumulating in

$Acsl4^{-/-}$ IECs after AA stimulation (Fig. 6c). A combination of EETs ameliorated AA-induced IL-6 and CXCL1 production in $siGpx4$ IECs (Fig. 6f, g), likely due to their anti-inflammatory effect that suppressed NF-κB[34]. In line with this, we noted that AA induced activation of NF-κB p65 in $siGpx4$ IECs (Supplementary Fig. 11I, J), and NF-κB inhibition with BAY11-7082 or MG132 abolished IL-6 production in $siGpx4$ IECs independent of LPO

**Fig. 4 Iron availability, lipoxygenases and LPO control PUFA-induced cytokine production. a** LPO quantification after 24 h stimulation with AA and ferric iron (5 μM Fe(III) sulfate) or vehicle (n = 6 biologically independent experiments). *P = 0.0234. **b, c** Quantification of IL-6 and CXCL1 in the supernatant from siGpx4 and siCtrl IECs stimulated with AA and ferric iron or vehicle for 24 h (n = 4 biologically independent experiments).*P = 0.0479 for (**b**) and *P=0.0259 for (**c**). **d, e** Quantification of IL-6 and CXCL1 in the supernatant from siGpx4 and siCtrl IECs stimulated with AA and deferoxamine (DFO) or vehicle (n = 6 biologically independent experiments). ***P < 0.001 for (**d**) and (**e**). **f** LPO quantification by flow cytometry of BODIPY581/591 C11+-labeled IECs after 24 h stimulation with AA and α-tocopherol (α-toco) or vehicle (n = 6 biologically independent experiments). *P = 0.0199. **g, h** Quantification of indicated cytokines in the supernatant from siGpx4 and siCtrl IECs after 24 h AA stimulation co-treated with LPO scavengers (n = 6 biologically independent experiments in (**g**) and n = 4 biologically independent experiments in (**h**)). ***P < 0.001. **i, j** Cytokine quantification in the supernatant of siGpx4 and siCtrl IECs after 24 h AA stimulation and treatment with selective LOX inhibitors (n = 5 for vehicle and n = 3 for indicated inhibitors; biologically independent experiments). *P < 0.05, **P < 0.01, ***P < 0.001. **k** Quantification of CXCL1 in the supernatant from siGpx4 and siAlox12 or siAlox15 co-silenced IECs stimulated with AA (20 μM) for 24 h (n = 4 biologically independent experiments). ***P < 0.001. Data are presented as boxblot with median and interquartile range (25th and 75th). The whiskers represent minimal and maximal values. One-way ANOVA with Bonferroni's multiple comparison test or Kruskal–Wallis Test with Dunn's multiple comparison test was used. Source data are provided as a Source Data file.

(Supplementary Fig. 11K, L). As such, Acsl4 deletion may limit AA-induced cytokine responses by modulating AA metabolism[34], while GPX4 controlled LPO.

## Discussion

Westernization of dietary habits, partially characterized by enrichment with PUFAs[35,36], paralleled the increased IBD incidence[19]. Previous observations associated PUFA uptake and mucosal AA accumulation with the risk of developing IBD[20,21,37]. Large prospective clinical trials in CD patients (and patients without IBD) indicated that PUFA supplementation may cause gastrointestinal side effects (e.g. diarrhea), indicative for disturbed intestinal homeostasis[38,39]. In contrast, dietary restriction (e.g. by an elemental diet) ameliorates the course of CD[40,41]. These reports and other studies[15] indicate that dietary cues impact the risk of developing CD and affect the natural history of disease. However, a direct link between PUFA uptake and intestinal inflammation remained elusive. Our study establishes that dietary-derived PUFAs trigger neutrophilic inflammation in the small intestine resembling some aspects of human CD.

IECs have the delicate task of maintaining a physical and immunological line of defense to protect the host against a potentially hostile environment. At the same time IECs must continue to allow uptake of essential nutrients such as long-chain fatty acids[42]. A hypoxic milieu and exposure to luminal noxae may specifically require GPX4 activity in IECs to protect against cellular LPO, a critical condition that determines cell fate[1]. While substantial advances have shaped our understanding of ferroptosis[1], GPX4-restricted immunologic responses remain poorly explored despite reports of inflammatory tissue injury in GPX4-deficient animals[9,11,27]. We report that IECs with reduced GPX4 activity respond to non-toxic dietary PUFA exposure with an inflammatory response involving IL-6 and CXCL1. PUFA-induced CXCL1 production of siGpx4 IECs was comparable to CXCL1 production induced by TNFα stimulation. In contrast IL-6 production was far less pronounced in PUFA-stimulated siGpx4 IECs, when compared to TNFα stimulation. Future studies will delineate the relevance of both cytokines as driver of mucosal inflammation in our model. Notably, exposing ω-3 PUFAs such as DHA (which is thought to exert anti-inflammatory effects[43]) elicited cytokine production similar to AA (which is thought to exert inflammatory effects) only in Gpx4-deficient IECs. As such it appears that the availability (rather than the positioning) of double bonds within PUFAs define their propensity to fuel LPO and an inflammatory response. These findings led us to explore the consequences of PUFA exposure on IECs with impaired GPX4 activity in the intestine. We generated mice with IEC-specific deletion of one Gpx4 allele (Gpx4+/−IEC), which is a valuable tool for modeling the effects of reduced, but not completely abrogated epithelial GPX4 activity as observed in patients

with small intestinal CD. Indeed, a PUFA-enriched WD containing 10% fish oil (with ω-3 and ω-6 PUFAs) evoked focal neutrophilic enteritis in male and female Gpx4+/−IEC mice that was characterized by granuloma-like mucosal to submucosal accumulation of inflammatory cells and expression of the IL-8 homologue Cxcl1. As such, our findings represent first evidence that non-toxic dietary lipids trigger focal enteritis in a genetically susceptible host resembling some aspects of human small intestinal CD. Future studies are warranted to explore the immune-phenotype in more detail. Notably, a WD rich in saturated fatty acids (without addition of fish oil) did not induce intestinal inflammation in Gpx4+/−IEC mice, indicating that PUFA supplementation specifically elicited the inflammatory phenotype.

We did not note colonic intestinal inflammation in PUFA WD- or AA/FM-exposed Gpx4+/−IEC mice, suggesting that environmental cues may determine disease localization. We speculate that PUFAs and iron are specifically absorbed in the small intestine[32,33,44], which could be one explanation for small intestinal disease localization and a requirement for GPX4 activity in the small intestine. In line with a small intestinal phenotype in mice, we specifically noted reduced epithelial GPX4 activity in active small intestinal CD. In contrast, we did not note altered GPX4 activity in colonic CD or UC.

Of note, fish oil supplementation has been tested in the maintenance of remission in CD patients with mixed results. More specifically, fish oil supplementation had favorable effects on maintenance of remission in some patients[45], while larger studies observed no beneficial effect but worsening of gastrointestinal symptoms (e.g. diarrhea)[45–47]. This observation indicates that dietary PUFA exposure impacts the course of CD. Similarly, PUFA supplementation induces gastrointestinal symptoms (diarrhea, abdominal pain and nausea) in patients without IBD[39]. Which factors define a beneficial or detrimental response on PUFA challenge (e.g. fish oil or other sources such as fish, meat and eggs) in CD (or healthy) patients is unknown. Our data indicate that PUFAs trigger GPX4-restricted mucosal inflammation resembling some aspects of human CD.

Ferroptosis is fundamentally controlled by lipoxygenase-driven LPO, which is limited by GPX4 and driven by cellular iron availability[1]. Similarly, we found that PUFA exposure induced LPO and cytokine production of siGpx4 IECs, while pharmacologic iron chelation with DFO, inhibition of lipoxygenases and LPO scavenging with α-tocopherol ameliorated this phenotype. Genetic deletion of Acsl4 (which is required for ferroptosis[2,6]), abrogated PUFA-induced cytokine production in IECs with reduced GPX4 activity. While GPX4 did not control AA metabolism, we found that ACSL4 limited the generation of anti-inflammatory AA metabolites (epoxyeicosatrienoic acids or 'EETs'), which have been demonstrated to inhibit NF-κB

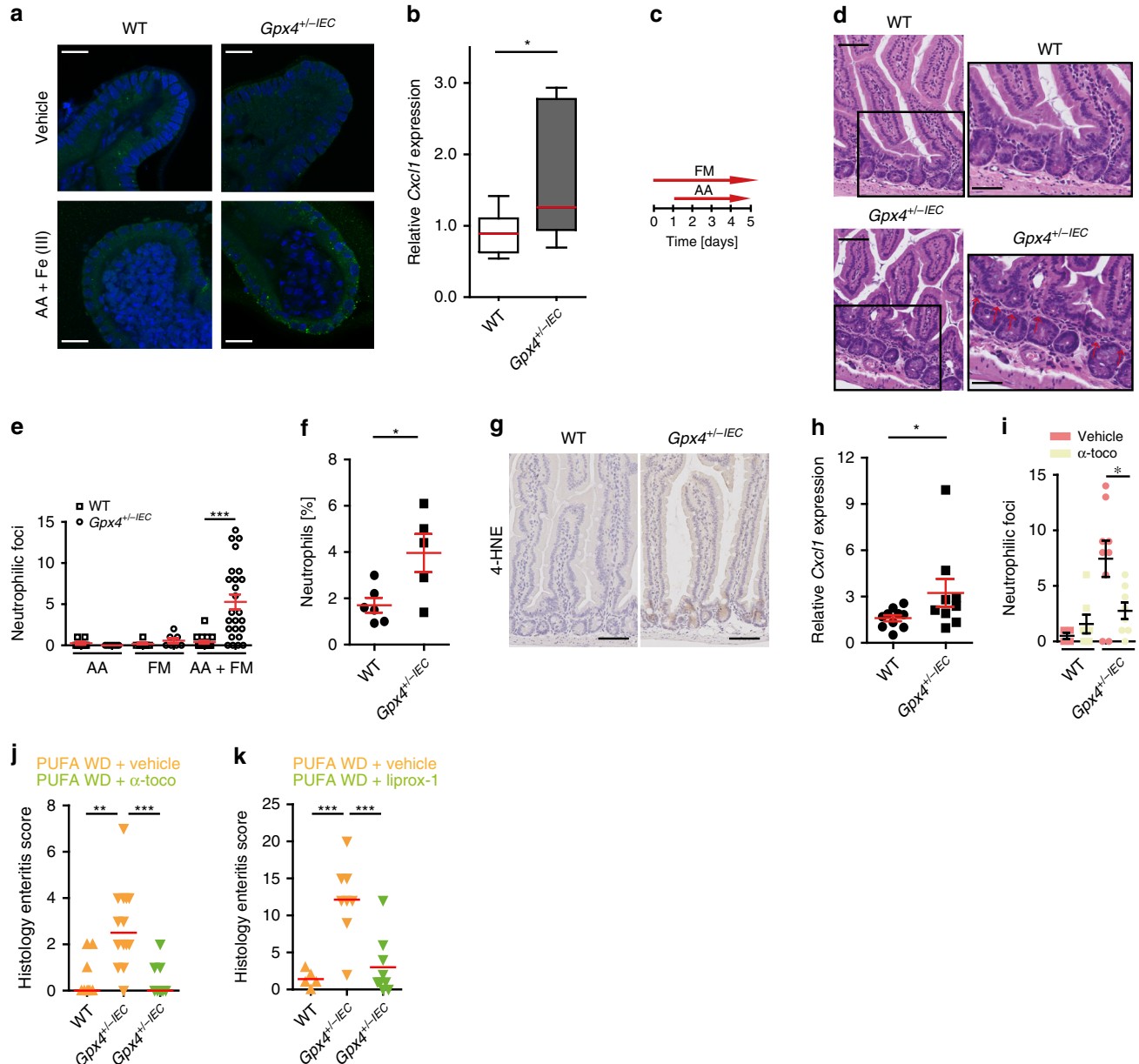

activity[34]. Indeed, a combination of EETs, but not a single EET alone, reduced AA-induced cytokine production similar to pharmacologic inhibition of NF-κB. These data suggest that GPX4-restricted LPO and ACSL4-controlled AA metabolism converge on NF-κB-mediated transcription of inflammatory cytokines specifically in GPX4-deficient IECs. As such, the same mecha-
nisms (i.e. iron availability, LOX-mediated LPO and ACSL4) that control ferroptosis[1] also control PUFA-induced cytokine pro-duction in *siGpx4* IECs. These findings lay the foundation for understanding dietary lipid-induced intestinal inflammation. Indeed, AA and ferric maltol (approved for the treatment of iron-deficiency anemia in IBD[31]) triggered epithelial LPO, *Cxcl1* expression and small intestinal neutrophilic infiltration in *Gpx4+/−IEC* mice. Similarly, *Gpx4+/−IEC* mice exposed to a PUFA WD exhibited increased LPO and neutrophilic inflam-mation, and LPO scavenging with α-tocopherol ameliorated enteritis in both models. These findings indicate that GPX4 protects against epithelial LPO, which sets a threshold for

intestinal inflammation triggered by dietary lipids. Although LPO is a key feature of ferroptosis[1], cell death was not a pre-requisite for intestinal inflammation indicated by a lack of epi-thelial TUNEL labeling in *Gpx4+/−IEC* mice. One reason for this may be that one *Gpx4* allele is sufficient to protect against fer-roptotic IEC death, but insufficient to prevent PUFA-induced cytokine production. In this context, studies in heterozygous *Gpx4* knockout mice did not report cell death (or related organ dysfunction)[11,29,48]. Future work may define a role for ferrop-totic cell death in mucosal inflammation.

Collectively, our data support a model in which dietary PUFAs elicit neutrophilic inflammation, which emanates from IECs with reduced GPX4 activity (Supplementary Fig. 12). Our findings turn the spotlight onto GPX4-restricted oxidative processes, which determine the competence of the epithelium to cope with environmental cues that inevitably occur on mucosal surfaces. Impaired GPX4 activity observed in small intestinal CD may arise from dietary long-chain fatty acids (Supplementary Fig. 5), (yet unidentified) immune mediators, bile salts[49] or microbial

**Fig. 5 AA and ferric maltol induce neutrophilic inflammation in $Gpx4^{+/-IEC}$ mice. a** Representative confocal microscopy images of 4-HNE-labeled organoids (green) from indicated genotypes after stimulation with AA and ferric iron or vehicle for 24 h. Scale bars indicate 20 µm ($n = 3$ biologically independent samples). **b** $Cxcl1$ expression after 48 h AA and ferric iron stimulation of indicated organoids determined by qPCR (n=11 biologically independent samples). Data are presented as boxplot with median and interquartile range (25th and 75th). The whiskers represent minimal and maximal values. *$P = 0.0303$. **c** Model of arachidonic acid and ferric maltol gavage. **d, e** Representative H&E images (**d**) and histology score (**e**) of indicated genotypes orally exposed to ferric maltol and/or AA as indicated in (**c**). The red arrows denote neutrophils. Scale bars indicate 100 µm and 50 µm, respectively. Each dot represents one experimental animal ($n = 8$ mice for WT AA, $n = 7$ mice for $Gpx4^{+/-IEC}$ AA, $n = 7$ mice for WT FM, $n = 7$ mice for $Gpx4^{+/-IEC}$ FM and $n = 18$ mice for WT AA+FM and $n = 25$ mice for $Gpx4^{+/-IEC}$ AA+FM). ***$P < 0.001$. **f** Neutrophilic infiltration of GR1⁺ neutrophils by flow cytometry of indicated genotypes from the experiment shown in (**c**). Each dot represents one experimental animal ($n = 6$ WT mice and $n = 5$ $Gpx4^{+/-IEC}$ mice). *$P = 0.0228$. **g** Representative images of 4-HNE immunoreactivity (brown), indicative for LPO ($n = 5$ mice per group). Scale bars indicate 100 µm. **h** Quantification of $Cxcl1$ expression determined by qPCR in intestinal epithelial scrapings of indicated genotypes from the experiment shown in (**c**). Each dot represents one experimental animal ($n = 10$ mice per group). $P = 0.0435$. Mann–Whitney test. **i** Histology score of AA- and ferric maltol-exposed mice with or without α-tocopherol supplementation [0.4 mg/ml] in drinking water over the course of the experiment. Each dot represents one experimental animal ($n = 4$ mice for WT + vehicle, $n = 7$ mice for WT + α-toco, $n = 9$ mice for $Gpx4^{+/-IEC}$ + vehicle and $n = 8$ mice for $Gpx4^{+/-IEC}$ + α⁻toco). $P = 0.0302$. **j** Enteritis histology score of WT and $Gpx4^{+/-IEC}$ mice exposed to a PUFA-enriched WD (PUFA WD) for 3 months with and without α-tocopherol supplementation [0.4 mg/ml] in drinking water over the course of the experiment. Each dot represents one experimental animal. Median shown ($n = 9$ mice for WT PUFA WD + vehicle, $n = 14$ mice for $Gpx4^{+/-IEC}$ PUFA WD + vehicle, $n = 9$ mice PUFA WD + α-toco). **$P < 0.01$. **k** Enteritis histology score of WT and $Gpx4^{+/-IEC}$ mice exposed to a PUFA-enriched WD (PUFA WD) for 3 months with and without liproxstatin-1 treatment intraperitoneally from 6 weeks [10 mg/kg] until the closure of the experiment. Each dot represents one experimental animal. Median shown ($n = 5$ mice for WT PUFA WD + vehicle, $n = 8$ mice for $Gpx4^{+/-IEC}$ PUFA WD + vehicle, $n = 9$ mice PUFA WD + liproxstatin-1). ***$P < 0.001$. For panel (**e**), (**f**), (**h**), and (**i**) data are presented as mean±SEM. For panel (**b**), (**f**) and (**h**) unpaired two-tailed Student's $t$ test and for panel (**e**) and (**i-k**) one-way ANOVA with Bonferroni's multiple comparison test was used. Source data are provided as a Source Data file.

metabolites[50] that are absorbed in the small intestine. As we specifically observed impaired GPX4 activity in IECs from the inflamed (but not uninflamed) mucosa of CD patients, it appears plausible that GPX4 deficiency is no primary defect in CD but occurs secondary to insults (likely caused by yet unidentified cellular stressors) in the inflamed mucosa. We speculate that IBD-related genetic cues may impinge on GPX4 activity during inflammation[51], and that genetic and environmental mucosal insults (other than PUFAs) trigger an inflammatory response in GPX4-deficient hosts[52]. Future studies will advocate a critical role of oxidative processes on cellular membranes of IECs and decipher the impact of oxidized lipid species in mucosal inflammation and IBD[53-55].

## Methods

**Human studies.** IBD patients were recruited in the gastroenterology outpatient clinic of the Medical University of Innsbruck and included when a definitive IBD diagnosis (by clinical, endoscopic, and histopathological means) was established and informed consent was obtained. Healthy controls undergoing screening colonoscopy—which lacked endoscopic and histological signs of intestinal disease—were included after informed consent was obtained. Patients were excluded if any of these criteria were not fulfilled or if the histology report and the endoscopic report conflicted. Ileal and colonic biopsies were collected from IBD patients and healthy controls, stored on ice in RPMI medium or formalin and processed the same day. One biopsy was used for mRNA isolation and one biopsy was formalin-fixed and paraffin-embedded for histology. Remaining biopsies were used to isolate IECs according to the section IEC isolation. IEC suspensions were snap-frozen in lysis buffer and stored at −80 °C until GPX4 enzymatic activity testing (see section GPX4 enzymatic activity assay) or protein analysis by western blot (see section Immunoblot).

**Mice.** C57BL/6J $Gpx4^{flox/flox}$ mice[27] were crossed with C57BL/6J $Villin$-$Cre^+$ mice[28] to obtain $Gpx4^{flox/wt};Villin$-$Cre^{+/-}$ ($Gpx4^{+/-IEC}$) mice or $Gpx4^{flox/flox};Villin$-$Cre^{+/-}$ ($Gpx4^{-/-IEC}$) mice. All experiments were performed with male and female 7- to 10-week-old mice. Littermates of the same sex were randomly assigned to experimental groups and were fed a chow diet or a Western diet (ssniff, TD88137) with or without 10% fish oil (ssniff supplementation, see Table S2). $Gpx4^{-/-IEC}$ breeding pairs were fed an α-tocopherol enriched diet (sniff, E157892-14, +500 mg/kg Tocopheryl Acetate) until weaning in the animal facility in Erlangen. In all other experiments $Gpx4^{+/-IEC}$ mice and WT controls were co-housed under SPF conditions (ZVTA) at the Medical University of Innsbruck.

**Mouse treatment.** 7- to 10-week-old WT and $Gpx4^{+/-IEC}$ mice received daily ferric maltol (0.4 mg, Shield Therapeutics PLC) and/or arachidonic acid (10 mg, Sigma, 10931) via oral gavage over the course of 6 days with or without α-tocopherol

supplementation (0.4 mg/ml, Sigma) in drinking water ad libitum. During the experiment, mice were fed a chow diet ad libitum. On day 6 mice were sacrificed for tissue collection.

Dextran sodium sulfate (DSS) colitis was induced in 8-week-old co-housed WT and $Gpx4^{+/-IEC}$ mice using 3% DSS (MP Biomedicals, 02160110) given in drinking water ad libitum for five consecutive days followed by tap water for the rest of the experiment. Disease activity was assessed as previously described[56].

To investigate the influence of a Western diet, 7- to 10-week-old WT and $Gpx4^{+/-IEC}$ mice were fed a Western diet (ssniff, TD88137) with or without 10% fish oil (ssniff supplementation, see Table S2) and compared to mice receiving a low-fat diet (ssniff, CD88137) for 3 months ad libitum with or without α-tocopherol supplementation (0.4 mg/ml, Sigma) in drinking water ad libitum. Bodyweight was assessed weekly. After 3 months mice were sacrificed for tissue collection.

To test the anti-inflammatory effect of Liproxstatin-1 7- to 10-week-old WT and $Gpx4^{+/-IEC}$ mice were fed a Western diet (ssniff, TD88137) with 10% fish oil (ssniff supplementation, see Table S2) for 3 months ad libitum. After 6 weeks of feeding, mice received 10 mg/kg Liproxstatin-1 (Cayman Chemicals, 17730) or vehicle (5% DMSO in PBS) every other day by i.p. injection until closure of the experiment.

**Blood count.** Heparin blood samples were used to perform complete blood count analysis on a Vet-ABC animal blood counter (scil animal care company GmbH, Viernheim, Germany).

**Cell culture.** MODE-K cells (*Mus musculus*, small IECs, kindly provided by D. Kaiserlian) were cultured in high-glucose DMEM (Lonza, BE12-604F), 10% FCS (Biochrome, S0115), HEPES (10 mM, Biochrome, L1613), non-essential amino acids (1 mM, Gibco, 11140-035), 100U/ml penicillin and 100 µg/ml streptomycin (1% Biochrome, 0257 F). HEK293T cells (*Homo sapiens*, female, embryonic kidney cells, ATCC, CRL-1573) were cultured in high-glucose DMEM (Lonza, BE12-604F), 10% FCS (Biochrome, S0115), HEPES (10 mM, Biochrome, L1613), non-essential amino acids (1 mM, Gibco, 11140-035), 100U/ml penicillin and 100 µg/ml streptomycin (1% Biochrome, 0257 F) and 2 mM sodium pyruvate (1% Biochrome, L 0473). Cells were cultured at 37 °C in 5% $CO_2$.

**Reagents.** The following reagents were used for cell stimulation: deferoxamine (0.1–20 µM, Sigma, D9533), ß-thujaplicin (Hinokitol, Sigma, 469521), deferasirox (Exjade®, Novartis), fe(III) sulfate hydrate (2-20 µM, Sigma, F0638), arachidonic acid (AA, 20 µM, Sigma, A3611), docosahexaenoic acid (DHA, 500 µM, Sigma, D2534), palmitic acid (PA, 250 µM, Sigma, P5585), oleic acid (OA, 250 µM, Sigma, O1008), palmitoleic acid (POA, 100 µM, Sigma P9417), stearidonic acid (SDA, 50 µM, Sigma SMB00291), eicosapentaenoic acid (EPA, 100 µM. Sigma E2011), docosapentaenoic acid (DPA, 25 µM, Sigma D1797) ferrostatin-1 (fer-1, 0.01–1 µM, Sigma, SML0583), α-tocopherol (α-toco, 0.1–1 µM, Sigma, T3251), Z-VAD-FMK (0.1–100 µM, BD Bioscience, 550337), TNFα (50 ng/ml, PeproTech), IFNγ (50 ng/ml, R&D), IL-1β (10 ng/ml, PeproTech), LPS (100 ng/ml, Sigma L4524), IFNβ (2500U/ml, PBL, 12401-1), IL-4 (10 ng/ml, PeproTech 214-14), IL-6 (20 ng/ml, PeproTech 216-16),

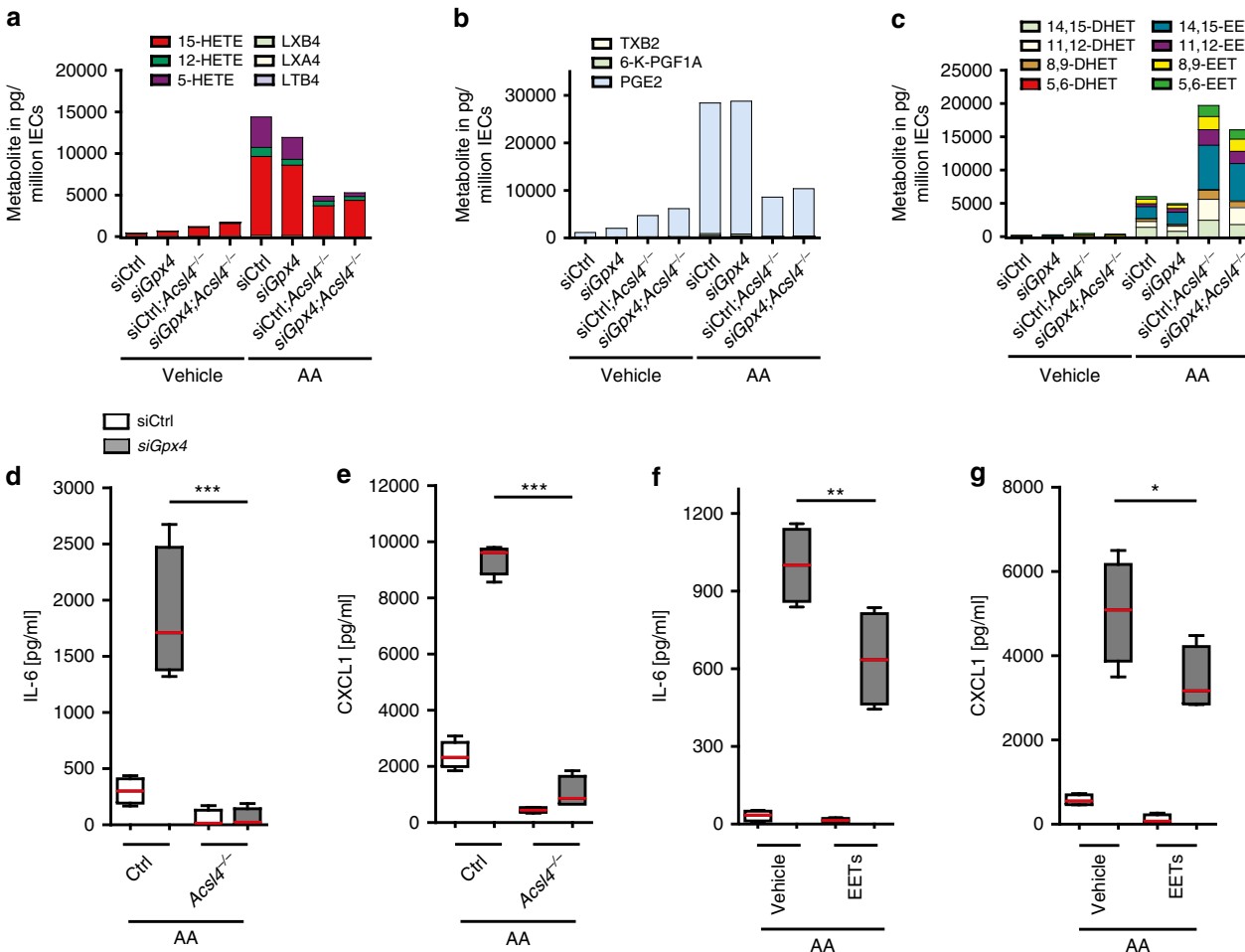

**Fig. 6 ACSL4 governs AA-induced cytokine production partly by modulation of AA metabolism. a–c** Quantification of LOX (**a**), COX (**b**), and P450 (**c**) AA metabolites in *siGpx4* and siCtrl IECs with or without deletion of *Acsl4* and AA stimulation for 24 h. Three independent experiments were performed and pooled for metabolite analysis by LC-MS/MS. **d, e** Quantification of IL-6 and CXCL1 in the supernatant of *siGpx4* and siCtrl IECs with or without deletion of *Acsl4* and AA stimulation for 24 h (n = 4 biologically independent experiments). ***$P < 0.001$. **f, g** Quantification of IL-6 and CXCL1 in *siGpx4* and siCtrl IECs after 24 h AA stimulation co-treated with a combination of EETs (0.25 µM of (+/−) 8(9)-EET, (+/−) 11(12)-EET and (+/−) 14(15)-EET) (n = 4 biologically independent experiments). **$P = 0.0011$ and *$P = 0.0242$. For panel (**d–g**) data are presented as boxblot with median and interquartile range (25th and 75th). The whiskers represent minimal and maximal values. One-way ANOVA with Bonferroni's multiple comparison test. Source data are provided as a Source Data file.

IL-22 (10 ng/ml, PeproTech 210-22), cholic acid (1 µM, Sigma C9282), deoxycholic acid (1 µM, Sigma D2510), ursodeoxycholic acid (1 µM, Sigma U5127),(+/−)8(9)- and (+/−)11(12)- and (+/−)14(15)-EETs (0.25 µM Cayman Chemicals, 50511, 50651, 50351), ML351 (10 µM, Sigma, SML1353), PD146176 (0.5 µM, Sigma, P4620), zileuton (10 µM, Sigma, Z4277), NCTT-956 (2 µM, Sigma, SML0499), piroxicam (20 µM, Sigma, P5654), 5-HETE (200nM-1µM, Cayman Chemicals, 34210), 15-HPETE (200nM-1µM, Cayman Chemicals, 44720), PGE2 (50 ng/ml, Sigma, P0409), BAY117082 (10 µM, Sigma, B5556), MG132 (125 nM, ApexBio, A2585).

**siRNA silencing**. MODE-K IECs were seeded on 6-well plates at ~70% confluence for siRNA silencing with either *Gpx4* siRNA (*siGpx4*, Ambion, s122098), *Alox12* (*siAlox12*, Ambion, s62261), *Alox15* (*siAlox15*, Ambion, s62271) or scrambled control siRNA (siCtrl, Ambion 4390843) and RNAiMAX (Thermo Fisher Scientific, 13778100) transfection over 48 h unless otherwise indicated, according to the recommended protocol.

**CRISPR Cas9 gene editing**. Target genes were disrupted in MODE-K IECs using the CRISPR/Cas9 system[26]. For each gene three guide RNAs (gRNA) targeting different exons of the target gene were designed. gRNAs with specific overhangs were annealed into a BsmBI-digested plentiCRISPRv2 plasmid (Addgene #52961, gift from Feng Zhang)[57]. Vectors were transfected into HEK293T cells (ATCC® CRL-1573™) to produce viral particles. Harvested supernatants were used for gene knockout. For transfection, MODE-K cells were seeded on 6-well plates and infected with viral particles containing the constructed vectors. The guide RNA sequence used for studies in Gpx4-/- IECs was CGTGTGCATCGTCACCAACG,

which targeted exon 1. The guide RNA sequence used for studies in *Acsl4*-/- IECs was CAATAGAGCAGAGTACCCTG, which targeted exon 6. To generate *Gpx4*-/- and *Acsl4*-/- clones by CRISPR Cas9 gene editing, transfection was performed for 48 h followed by puromycin (Gibco, A1113803) selection for 10 days and subsequent seeding onto 96-well plates with one cell per well for expansion.

**Human intestinal epithelial organoids**. Human organoids were cultured from IEC isolates of biopsy specimens retrieved from endoscopy of CD patients and healthy controls using IntestiCult Organoid Growth Medium (Stemcell Technologies) and a protocol adapted from the manufacturer's instructions. Briefly, biopsies were flushed with 10 ml of ice cold PBS and minced into the small pieces. Tissue was then transferred to 5 ml Gentle Cell Dissociation Reagent (Stemcell Technologies) and incubated at 4 °C on a rocking platform for 30 min. After centrifugation at 4 °C and 290*g* for 5 min supernatant was removed and crypts were transferred to 1 ml of ice cold 1% BSA/DMEM. Crypts were dissolved by gentle mixing and passed through a 70 µm cell strainer. Seeding was identical to that for mouse organoids (see below). Human organoids were passaged with a split ratio of 1:3 every 7–14 days.

**Mouse intestinal epithelial organoids**. Intestinal epithelial organoids were cultured from 4- to 8-week-old *Gpx4*+/−*IEC* and littermate WT mice using IntestiCult Organoid Growth Medium (Stemcell Technologies, 06005) and a protocol adapted from manufacturer's instructions as initially described[58]. Briefly, small intestines were flushed with ice cold PBS, minced to pieces of approximately 2–3 mm in size and washed up to five times with 10 ml ice cold PBS. Samples were transferred to

2 mM EDTA/PBS and incubated at 4 °C on a rocking platform for 30 min. After sedimentation supernatant was removed and crypts eluted in 10 ml PBS by shaking vigorously and passing crypts through a 70 μm cell strainer to obtain Fraction 1. This process was repeated three times to obtain Fractions 2–4. Fractions were analyzed under a light microscope and the optimal fraction was chosen to obtain crypts for organoid culture by centrifugation at 290g for 5 min at 4 °C. Crypts ($N = 500$) per well were seeded in 50 μl Matrigel (BD, 356231) on a pre-warmed 24-well plate and allowed to solidify for 10 min at 37 °C, after which 500 μl IntestiCult Growth Medium supplemented with 100U/ml penicillin and 100 μg/ml streptomycin (Biochrome, 0257F) was added. Medium was exchanged three times per week and organoids passaged with a split ratio of 1:6 every 7–14 days (Supplementary Fig. 8A-G).

**Stimulation of mouse organoids**. Before stimulation, organoids were allowed to establish for 6 days. Organoids were stimulated with arachidonic acid (AA, 100 μM, Sigma, A3611) and Fe(III) sulfate (5 μM, Sigma, F0638) or vehicle for indicated time periods depending on the experiment setting. Medium was replaced every 24 h. For RNA extraction TRIzol reagent (Invitrogen, 15596026) was used according to the manufacturer's instructions.

**Immunofluorescence Imaging**. Organoids were cultured with 15 μl Matrigel (BD, 356231) on chamber slides (Falcon, 354108) covered with 200 μl IntestiCult Growth Medium supplemented with 100U/ml penicillin and 100 μg/ml strepto-mycin (Biochrome, 0257 F). Medium was exchanged three times a week and organoids were allowed to establish for 6 days. Organoids were stimulated as described above for 24 h, washed with PBS and fixed with 4% PFA at room temperature for 20 min, washed again twice with PBS and permeabilized with PBS-Triton X 0.5% for 20 min at room temperature. Organoids were washed with IF buffer (PBS, Triton X 0.3%, Tween 0.05%), blocked with blocking solution (PBS-Triton X 0.3%, Tween 0.05%, BSA 1%), followed by another wash with IF buffer and primary antibody in blocking solution was added over night at 4 °C. Organoids were washed three times with IF buffer and secondary antibody was added for 1 h, washed three times with IF buffer, mounted (Invitrogen, P36962) and analyzed with a Zeiss Axio Observer Z1 confocal microscope and Zen 2012 software. Intestinal sections were cryopreserved in O.C.T., acetone-fixed, stained with anti-GR-1 and analyzed with the confocal microscope stated above.

The following antibodies were used for immunofluorescence: anti-4HNE (1:200, Abcam, ab46545), anti-GPX4 (1:400, Abcam, ab125066) and anti-GR-1 (1:100, BioLegend, 108413) as primary antibodies and anti-rabbit IgG Alexa Fluor 488 (1:1000, Invitrogen, A-11034) and anti-rat IgG Alexa Fluor 594 (1:1000, Invitrogen, A-11007) as secondary antibody.

**Histology**. Formalin-fixed paraffin-embedded sections were hematoxylin & eosin (H&E)-stained and assessed for inflammation by an expert pathologist using a light microscope (Zeiss, Germany). Images were captured with a Zeiss AxioCam. In AA and FM gavage experiments, acute inflammation was quantified by determining the number of neutrophilic spots (>3 neutrophils in one visual field on ×20 magnification) per section. We did not note mononuclear infiltration, hyperplasia or ulceration in our AA and FM exposure experiments. DSS colitis severity and enteritis evaluation was performed as reported[59]: Briefly, a semi-quantitative scoring system was used that was composed of five histological subscores. Histo-logical subscores (0, absent; 1, mild; 2, moderate; 3, severe) were mononuclear cell infiltration, crypt hyperplasia, epithelial injury/erosion, polymorphonuclear cell infiltration and transmural inflammation. The sum of these subscores was multi-plied by a factor that reflected the extent of inflammation along the intestine (1, 10%; 2, 10–25%; 3, 25–50%; and 4, >50%).

**Immunohistochemistry, TUNEL, BrdU, and PAS labeling**. Formalin-fixed par-affin-embedded sections were deparaffinised, rehydrated, antigen-retrieved for 15 min in sodium citrate at subboiling temperatures and peroxidase-blocked. The primary antibody was incubated over night at 4 °C and the secondary biotinylated antibody mediated horseradish peroxidase (HRP)-driven 3,3′-diaminobenzidine (DAB, DAKO, K3468) turnover, which resulted n brown labeling of immunor-eactive cells. Stained sections were analyzed with a light microscope (Zeiss, Ger-many) and captured with a Zeiss AxioCam. TUNEL labeling was performed according to the manufacturer's instructions (Roche, 11684817910). TUNEL-positive cells were quantified in 50 consecutive crypts and depicted as TUNEL⁺ cells of total IECs. Proliferation of IECs was investigated by intraperitoneal injection of 5-bromodeoxyuridine (BrdU, BD Pharmigen, 550891) and detected after 24 h using a BrdU in situ detection kit (BD Pharmigen, 550803). BrdU⁺ cells of total IEC along the villus-crypt axis were analyzed. PAS reaction was performed according to a standard protocol and PAS⁺ cells were counted in 50 crypt/villus axes.

The following antibodies were used for immunohistochemistry: anti-GPX4 (1:400, Abcam, ab125066), anti-4HNE (1:400, Abcam, ab46545) and anti-MPO (1:200, Dako, IS511) were employed with a secondary biotinylated antibody (Vector, MP-7401).

**Mouse IEC isolation**. Mouse IEC isolation protocol was performed as previously described[59]. Briefly, PBS-flushed and longitudinally cut intestinal pieces were vortexed in ice cold PBS for 5 min, transferred to 30 mM EDTA, and vortexed for 5 min. The supernatant was collected and the procedure was repeated for a total of four times. Supernatants were microscopically checked for crypt IEC enrichment and then spun down at 800g for protein extraction or flow cytometry labeling.

**Human IEC isolation**. The biopsies were moved from RPMI (Biochrome, FG1385) to HBSS-CMF buffer (Gibco, 14175-053, 0.5% BSA, 2 mM EDTA and DTT). Samples were incubated on a shaker for 20 min at room temperature. The samples were then vortexed vigorously and supernatant (containing IECs) was collected through a 100 μm cell strainer, which was repeated for a total of three times. Supernatant was then spun down at 300g and the pelle t was used for GPX4 activity assay and western blot analysis as detailed in the respective sections.

**Murine LPMC isolation**. Mouse LPMCs were isolated according to previously published protocols[60,61]. In short, the proximal small intestine was flushed with ice cold PBS, opened longitudinally, cut into small pieces and transferred to HBSS (Gibco, 14175-053) containing 10% FCS, DTT (1 mM) and EDTA (2 mM) fol-lowed by shaking for 20 min at room temperature. Samples were vortexed to remove IELs and the tissue was washed and collected in IMDM (Gibco, 21056-023) 20% FCS. Tissue was washed and 10U/ml DNAse (10U/ml, Sigma, D8764) and 128 U/ml collagenase (128U/ml, Sigma, C1889) digested on a shaker for 60 min at 37 °C. Cells were passed through cell strainers (100 μm) and washed twice before transferring to cytometry buffer for staining (see below).

**LPO and cell death labeling**. Cells derived from cell culture or from IEC isolation procedures were incubated with BODIPY 581/591 C11 or the surface labeling antibodies (see below) at 37 °C in the dark for ten to 30 min in flow cytometry buffer (2% FCS, 2 mM EDTA in PBS). Cells were subsequently washed with PBS, resuspended in FACS buffer and transferred through a 40μm cell strainer for flow cytometry. Annexin V, PI or 7AAD were used for cell death analysis.

**Flow cytometry analysis**. For LPO analysis, BODIPY-positive cells among DAPI-negative cells were analyzed as compared to a control sample using BODIPY measurement. For cell death analysis, debris was excluded using FSC/SSC char-acteristics; Annexin V and PI or 7AAD positivity was determined by flow cytometry.

**FACS gating strategy**. The gating strategy for analyzing the mucosal cellular infiltrate is depicted in Supplementary Fig. 9A, E. Briefly, cells were gated using FSC/SSC characteristics. Singlets were selected by comparing FSC width and FSC area. Neutrophils were identified as CD45⁺, Lin1⁻ (Lin1=CD3, CD19, CD49b, DAPI) and GR1⁺ cells. Macrophages were identified by CD45⁺, Lin1⁻, GR1⁻, CD11b⁺, MerTK⁺. Monocytes were characterized by CD45⁺, Lin1⁻, GR1⁻, Ly6Cʰⁱ. Dendritic cells were characterized by CD45⁺, Lin1⁻, GR1⁻, CD11c⁺ and MHCII⁺. T helper cells were identified by Lin2⁻ (Lin2=CD11c, F4/80, GR1, DAPI), CD3⁺, CD19⁻, CD4⁺. Cytotoxic T cells were identified by Lin2⁻, CD3⁺, CD19⁻, CD8⁺. B cells were defined as Lin2⁻, CD3⁻, CD19⁺. Details of antibodies used are found in Supplementary Table 4.

**RNA extraction and qRT-PCR**. RNA was isolated from 6-well plates, epithelial scrapings or human intestinal biopsies using an RNeasy mini kit (Quiagen 74104). RNA was transcribed into cDNA using M-MLV reverse transcriptase (Invitrogen 28025013). Quantitative real time PCR was performed with SYBR green mastermix (Eurogentec RT-SY2X-06+WOULR) on a MX3005 Stratagene cycler (Agilent). The primer used in this study can be found in Supplementary Table 4.

**Immunoblot**. Western blot analysis was performed according to standard protocols (Bio-Rad Laboratories). Briefly, isolated cells (from culture plates or scrapings) were lysed in RIPA buffer (50 mM Tris, pH 7.4, 150 mM NaCl, 1% Nonidet P-40, 0.5% sodium deoxycholate, and 0.1% SDS) or M-Per (Thermo Fisher Scientific, 78501) and supplemented with protease and phosphatase inhibitors (Thermo Fisher Scientific, 78443). Protein quantity was determined by Bradford assay (Bio-Rad Laboratories, 5000006) and equal amounts of protein were denatured at 95 °C in Laemmli buffer, resolved on SDS-PAGE and transferred to a polyvinylidene fluoride membrane (Sigma, GE10600023). After blocking the membrane in 5% skim milk, primary antibody was incubated over night at 4 °C. Signal was visualized with HRP-conjugated secondary antibodies (Cell Signalling Technology, 7074) and ECL Select Western Blotting Detection Reagent (Amersham, RPN2235). Densi-tometry of immunoblots was performed with ImageJ. All uncropped and unedited blots are available in the Source Data. The following antibodies were used: anti-GPX4 (1:2000, Abcam, ab125066), anti-ACSL4 (1:1000, Abcam, ab155282), anti-TfR (1:1000, Invitrogen, H68.4), anti-FPN1 (1:1000, Eurogentec, Lìege, Belgium) anti-ferritin (1:1000, Sigma, F5012), anti-phospho-NF-κB p65 (1:1000, Cell Sig-nalling Technology, 3039), anti-NF-κB p65 (1:1000, Cell Signalling Technology, 8242), with anti-GAPDH (1:2000, Cell Signalling Technology, 2118) or anti-ß-actin (1:2000, Sigma, A2066 and Abcam, ab49900) as loading control.

**Cytokine quantification**. Cellular supernat ants were collected, centrifuged at 300*g* for 5 min and stored at −20 °C. Cytokine quantification was performed by ELISA (IL-6, BD Biosciences, 555240; CXCL1/KC, R&D, DY453) according to the manufacturer's protocol.

**Cell viability assay**. Cells were seeded on 96-well plates (2000 cells per well) and treated with siRNA for 48 h. Cell viability was assessed by AlamarBlue turnover (Thermo Fisher Scientific, DAL1025) according to the manufacturer's recommendations.

**Quantification of iron uptake and release**. Iron uptake and release were performed as previously described[62]. Briefly, MODE-K IECs were silenced for 48 h with *siGpx4* or siCtrl and washed with high-glucose DMEM (1% FCS, 1% penicillin/streptomycin, 25 mM HEPES). To determine non-transferrin-bound iron uptake and release, cells were incubated with 5 µM [59] ferric chloride (Perkin Elmer, NEZ037) for 2 h. After washing, cells were transferred to high-glucose DMEM (1% FCS, 1% penicillin/streptomycin, 25 mM HEPES) and incubated for 1 h. Iron uptake and release were measured with a γ counter (Perkin Elmer). Uptake and release in CPM (counts per minute) were normalized to protein quantity as determined by BCA Assay (Thermo Fisher Scientific, 23225).

**GPX4 enzymatic activity assay**. Cells were collected in lysis buffer (100 mM Tris pH 7.6, 5 mM EDTA, 1 mM NaN₃ and 0.1% peroxide-free Triton-X100). Lysates were complemented with 0.6 U/mL glutathione reductase (Sigma, G3664), 0.2 mM nicotinamide adenine dinucleotide phosphate hydrogen (NADPH, Sigma, N7505), 3 mM reduced glutathione (GSH, Sigma, G4251) and 200 µM of the substrate cumene hydroperoxide (CHP, Sigma, 247502). NADPH turnover was measured on an Infinite 200PRO reader (Tecan) at 340 nm over 10 min at 37 °C. Enzymatic activity was calculated after subtracting absorbance decay obtained from buffer without cell lysates by using NADPH extinction coefficient of 6220/M/cm and by normalizing to total protein content[63].

**LC-MS/MS analysis**. The extraction protocol and analysis of bioactive lipids by LC-MS/MS were performed by Ambiotis SAS (Toulouse, France). Briefly cell pellets were supplemented with methanol and spiked with deuterated internal standards (ISTD) (PGE2-d4, 5-HETE-d8, LtB4-d4, LXA4-d5). After standing at −20 °C for 1 h and centrifugation, supernatants were diluted with H2O and subjected to solid phase extraction (SPE) on an HLB 96 well plate (Oasis). Methyl formate-eluted lipid mediators were concentrated by a nitrogen stream evaporator before LC-MS/MS analysis. LC-MS/MS experiment was performed on a 1290 Infinity U-HPLC system (Agilent Technologies, Santa Clara, CA, USA) mounted with a Kinetex Biphenyl column (2.1 mm, 50 mm, 1.7 µm, Phenomenex), maintained at 50 °C. The U-HPLC system was coupled to a 6490 triple quadrupole MS (Agilent Technologies), equipped with electrospray ionization source, performed in negative ion mode. Analyses were performed in multiple reaction monitoring detection mode by use of nitrogen as collision gas. Peak detection, integration, and quantitative analysis were done by use of MassHunter Quantitative Analysis Software (Agilent Technologies). Results are expressed in a quantitative manner (i.e. pg/sample). Validation and extraction efficacy of the extraction protocol was validated in Le Faouder et al[64]. Data was deposited in Mendeley (https://data.mendeley.com, 10.17632/k9ync2kd3g.2; Data for: Dietary lipids fuel GPX4-restricted enteritis resembling CD).

**Statistics**. Data are expressed as mean±standard error of mean (SEM) for all in vivo experiments. Data derived from in vitro experimentation is presented as box and whisker plot with median and first and third quartile (boundaries). The whiskers represent minimal and maximal values. Results shown are of at least three independent experiments with two technical duplicates, unless stated otherwise. Statistical significance was tested using an unpaired two-tailed Student's *t* test, a Mann–Whitney U test, a one-way or two-way ANOVA with Bonferroni correction or a Kruskal–Wallis test followed by Dunn's correction as indicated in the figure legend. Significance was considered $P < 0.05$. Grubbs' Test allowed determination and exclusion of one significant outlier in a parametric sample set. Graph Pad Prism version 5.04 was used for statistical analysis.

**Study approval**. We confirm that we comply with all relevant ethical regulations regarding the use of research animals and human study participants. Human studies were approved by the Ethics Committee of the Medical University of Innsbruck (UN4994) and informed consent was obtained prior to sample collection. Mouse experiments were performed in accordance with institutional guidelines of the Medical University of Innsbruck and following approval by federal authorities (BMWFW-66.011/0061-WF/V/3b/2016, BMBWF-66.011/0085-V/3b/2018 and BMBWF-66.011/0160-V/3b/2019 (Innsbruck); 55.2.2-2532-2-1043 (Erlangen)).

**Reporting summary**. Further information on research design is available in the Nature Research Reporting Summary linked to this article.

## Data availability

The data of this study are available from the corresponding author upon reasonable request. The dataset generated in this study are deposited in a publicly available platform (https://data.mendeley.com, 10.17632/k9ync2kd3g.2; Data for: Dietary lipids fuel GPX4-restricted enteritis resembling Crohn's disease). The source data for Figs. 1–6 and Supplementary Fig. S1-11 are provided in the Source Data File.

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

## Acknowledgements
The authors are grateful for the support received from the Austrian Science Fund (FWF, P 29379-B28), the Tyrolean Science Fund (TWF, 0404/1812), the Austrian Society of Gastroenterology and Hepatology (ÖGGH) and the European Crohn's and Colitis Organisation (ECCO) (to T.E.A.). We thank the Excellence Initiative (Competence Centers for Excellent Technologies—COMET) of the Austrian Research Promotion Agency FFG: Research Center of Excellence in Vascular Ageing Tyrol, VASCage (K-Project Nr. 843536) funded by BMVIT, BMWFW, Wirtschaftsagentur Wien and Standortagentur Tirol for their financial support (to H.T.). This work was further supported by the German Funding Agency (DFG) through the CRC1182 (C2 to F.S. & P.R.), the excellence initiative EXC306 (to P.R.) and EXS2167 (to P.R.). We thank the German Funding Agency (DFG) through TRR241 (A03 to C.B.) and SFB1181 (C05 to C.B.). We appreciate the support of the Christian Doppler Research Foundation and the Austrian Federal Ministry of Science, Research and Economy and the National Foundation for Research, Technology and Development (to A.R.M and G.W.). We thank Cornelia Wiedner Stiftung and Deutsche Arbeitsgemeinschaft für chronisch entzündliche Darmerkrankungen (DACED) for financial support (to T.E.A). Finally, we thank Karina Greve and the IKMB NGS lab for excellent technical support.

## Author contributions
H.T. and T.E.A. contributed equally to this work. L.M. and F.G. designed, performed, and analyzed most experiments and helped prepare the manuscript together with T.G., J.S., I.R. L.N., G.W.H., B.R., K.K., N.P., P.T., R.H., D.H., M.S., C.F., R.R.G., and B.E. G.O. and S.S. provided pathology expertise. M.K. helped with LC-MS/MS methodology. Q.R. and C.B. provided an experimental tool. R.K., M.E., and H.Z. provided access to clinical samples. F.S., P.R., H.Z., A.R.M., I.T., G.W., and A.K. helped develop the project. H.T. and T.E.A coordinated the project and prepared the manuscript.

## Competing interests
The authors declare no competing interests.
