## [Peer Review File · Nature Communications]

Reviewers' Comments:

Reviewer #1:

Remarks to the Author:

In this manuscript the authors present evidence that Crohn's disease ileal epithelial cells exhibit reduced levels of glutathione peroxidase 4 (GPX4) and are thus more susceptible to injury due to lipid peroxidation. As a result, mice with reduced GPX4 levels in intestinal epithelial cells manifest inflammation of the small intestine when fed a diet enriched in PUFAs (polyunsaturated fatty acids). In addition, epithelial cells with reduced GPX4 expression produce increased amounts of IL-6 and CXCL1. These data provide a mechanism for the possible role of certain PUFAs in the pathogenesis of Crohn's disease. However, it should be noted that attempts to treat patients with diets that contain reduced levels of omega-6 PUFAs have yielded inconsistent results and the positive results obtained have been marginal.

Comments:

1. GPX4 activity in non-lesional CD was not impaired. This argues that reduced GPX4 levels is not a primary CD defect. GPX4 is not impaired in lesional colonic epithelial cells; how and why do these cells differ from lesional ileal epithelial cells?
2. In figure 2 both omega-3 and omega-6 PUFA's elicit cytokine/chemokine responses from GPX4-deficient IEL. Shouldn't there be a difference inasmuch as omega-6 PUFAs are said to have harmful effects whereas omega-3 PUFAs are said not to have such effects.
3. Does exposure of epithelial cells to pro-inflammatory cytokines such as TNF-a or IFN-g have effects on GPX4 levels in such cells.
4. Mice with reduced GPX4 levels in epithelial cells have increased severity of DSS-colitis. This does not correspond to the fact that colonic cells in humans with have normal GPX4 levels and no inflammation of the colon was found in mice with reduced epithelial GPX4 levels on a PUFA-enriched diet. What accounts for this discrepancy?
5. In Figure 3, neutrophils and other parameters of inflammation in WT mice not on a PUFA-enriched diet should be included to show if the diet elicits inflammation even in the absence of GPX4 deficiency.
6. The detrimental effects of PUFA's in the GPX4-deficient mice is indeed contrary to what one might expect given the effect of PUFAs on the course of disease in humans, as the authors point out. It might be useful to test the effect of PUFAs in several strains of GPX4-deficient mice to see if there are strain differences in susceptibility to PUFA -induced injury in the face of GPX4 deficiency.

Reviewer #2:

Remarks to the Author:

The submitted article, "Dietary lipids fuel GPX4-restricted enteritis resembling Crohn's disease" described the role of intestinal epithelial glutathione peroxidase 4 in controlling the gut homeostasis.

Abstract: Expand the gene name ACSL4. In the key words, is acronym missing for the intestinal inflammation?

Introduction: The title is appropriate with a clear focus on dietary lipid-induced intestinal inflammation Crohn's disease. Authors have thoroughly reviewed the relevant literature. However, the importance of biolipid analysis and analytical techniques used in the detection and

determination of lipids should be added.

LC-MS/MS Methodology:

1. More information is needed for the extraction and quantification methods used for LC-MS/MS lipid analysis. A short method description in the supplemental data should be provided.
2. The authors did not state clearly the results were semi or absolute quantitation of lipids. As the two references that were provided showed that the method is semi-quantitative analysis of lipids, while Ambiotis SAS (Toulouse, France) lipidomics core facility claims absolute quantification of lipids on their webpage.
3. According to the reference 59, the method is fully validated with necessary experiments. There is no clarity that the authors have reproduced all the experiments and results or some part. A brief description about the validation experiments is needed including the results.
4. The authors need to clarify on the purpose of using Ambiotis SAS (Toulouse, France) standard operating procedure in the analyses. Provide details on the parameters or part used from this procedure.
5. The authors have not clarified on the type of samples, number of biological replicates used per group, names and concentrations of complete list of analytes (lipids). Provide the list as a supplementary table.
6. LC-MS/MS raw data should be deposited in public repositories e.g., MetaboLights etc.

Reviewer #3:

Remarks to the Author:

In this work, the authors studied how lipid peroxidation which is efficiently controlled by glutathione peroxidase 4 (GPX4) may impact on Crohn's disease (CD), a form of inflammatory bowel disease. This study is of particular relevance given the reported association of a single nucleotide polymorphism within GPX4 and CD. The authors now show that intestinal epithelial-specific ablation of Gpx4 in mice subjected to PUFA-enriched Western diet develop some form of neutrophilic enteritis which can be compensated by dietary vitamin E supplementation. Importantly, the authors also provide evidence that CD patient biopsies display reduced GPX4 activity and markers of lipid peroxidation. Overall, this is a very interesting and well-performed study providing mechanistic insights into the role of GPX4 in the development of CD.

Comments:

- Page 6: The authors did not obtain intestinal epithelia-specific homozygous Gpx4 null mice as these mice seem to die during embryogenesis – did the authors consider to rescue embryonic lethality by supplementing the diet with high vitamin E. What is the actual vitamin E content of the diet used in light of various papers showing that some KO models for GPX4 can be actually rescued by high vitamin E concentration. It may worth trying but at least it should be discussed here.
- Figure 3: Did the authors try to rescue diet-induced focal enteritis with cognate ferroptosis inhibitors such as ferrostatin-1 or liproxstatin-1 similar to what was done with alpha-tocopherol supplementation in fig. 5? Would leukocyte/neutrophil infiltration be sensitive to such a treatment?
- Figure 4: the authors try to make a link between lipoxygenase activity and ferric iron induced cytokine responses. As most "lipoxygenase-specific" inhibitors display non-specific radical trapping activity (RTA) (!) one may wonder whether there is an actual contribution of lipoxygenase(s) to the findings presented here. As long as the authors do not rule out that these compounds harbor RTA activity (see Shah et al ACS Cent Sci 2018), it cannot be stated here (see also page 12, last paragraph). Otherwise data on knockdown/knockout of one or several lipoxygenase(s) should be included here.
- Figure 5: the authors may indicate how much alpha-tocopherol was supplemented in the drinking water? Would ferroptosis inhibitors confer similar effects (see my comment in the foregoing)?
- For the general reader, it might be appropriate to include a model illustrating the main effects described here.

Minor:

- Page 3, first paragraph: the first reports showing that GPX4 protects from lipid peroxidation and non-apoptotic cell death are Ursini et al. 1982 and Seiler et al. Cell Metab 2008, respectively.
- Page 5, it should read Figure 2F and Figure 2G,H.

Point-to-point response NCOMMS-19-16671

Reviewer #1

“In this manuscript the authors present evidence that Crohn's disease ileal epithelial cells exhibit reduced levels of glutathione peroxidase 4 (GPX4) and are thus more susceptible to injury due to lipid peroxidation. As a result, mice with reduced GPX4 levels in intestinal epithelial cells manifest inflammation of the small intestine when fed a diet enriched in PUFAs (polyunsaturated fatty acids). In addition, epithelial cells with reduced GPX4 expression produce increased amounts of IL-6 and CXCL1. These data provide a mechanism for the possible role of certain PUFAs in the pathogenesis of Crohn's disease. However, it should be noted that attempts to treat patients with diets that contain reduced levels of omega-6 PUFAs have yielded inconsistent results and the positive results obtained have been marginal.”

We thank the reviewer for the positive summary of our work. We appreciate the notion that withdrawal of “omega-6 PUFAs have yielded inconsistent results”. Allow us to summarize dietary intervention studies in CD:

1. We agree that clinical data on the benefits of ω -3 PUFAs in IBD are still controversial and conflicting, especially in Crohn's disease (CD) as recently described in a Cochrane review ¹. For example, fish oil supplements (which contain ω -3 and ω -6 PUFAs) have been tested in the maintenance of remission in CD patients with mixed results. Fish oil supplementation had favourable effects for the maintenance of remission in 39 CD patients ², while larger scale studies (EPIC-1 and EPIC-2) observed no benefit but worsening of gastrointestinal symptoms (e.g. diarrhea) which may reflect worsening of CD ^{1,3}. These observations indicate that dietary PUFA exposure impacts the *course* of CD. Which factors define a beneficial or detrimental response to a PUFA challenge (e.g. fish oil or other sources such as fish, meat and eggs) in CD patients is unknown. Our data demonstrate that epithelial GPX4 critically restricts an inflammatory response in the intestine that is *triggered* by PUFAs. It may be tempting to speculate that reported inconsistencies (in heterogeneous study groups) arose consequent to the competence of the intestinal epithelium to cope with environmental cues that are restricted by GPX4. Whether this is indeed true for the development *or* course of Crohn's disease requires further clinical studies. We respectfully suggest that this is beyond the scope of our work. Notably, two recent large prospective randomized control trials did not report a benefit of ω -3 PUFAs on the course of cardiovascular disease⁴ or the development of colorectal cancer⁵. However, common gastrointestinal side effects (diarrhea, abdominal pain, nausea) were found in up to 45% of patients (even without inflammatory gastrointestinal disease)⁵. Collectively, these studies indicate that ω -3 PUFAs may affect intestinal homeostasis in healthy and in CD patients.

2. We acknowledge that dietary intervention studies in IBD are only in their infancy⁶, which is why we believe that our study is of importance. We respectfully disagree that dietary interventions have little impact on the course of IBD. Specifically, an elemental diet may effectively induce remission in CD patients to a similar extent as prednisolone⁷ which is associated with reduced expression of IL-6 and IL-8 ⁸. Moreover, an exclusive enteral nutrition is similarly effective as corticosteroids in pediatric CD patients⁹. These dietary schemes partly reduce the ω -3 and ω -6 PUFA content in the diet but are challenging to adhere⁹. For example, 41% percent of Crohn's disease patients receiving an exclusion diet showed improvement in clinical activity (compared to 16% controls). Efficacy in this study was attributable to the exclusion of pork, beef, eggs and milk, potent sources of polyunsaturated fatty acids¹⁰.

We thank the reviewer for highlighting this issue and we now discuss these aspects in more detail in

the revised manuscript.

“1.GPX4 activity in non-lesional CD was not impaired. This argues that reduced GPX4 levels is not a primary CD defect. GPX4 is not impaired in lesional colonic epithelial cells; how and why do these cells differ from lesional ileal epithelial cells?”

We appreciate this careful observation that also raised our interest. Indeed, GPX4 activity is impaired in IECs derived from the lesional mucosa of CD patients when compared to healthy controls. In contrast, GPX4 activity in IECs derived from the non-lesional mucosa of Crohn’s disease or lesional mucosa of ulcerative colitis patients was comparable to that of healthy controls. These observations suggest that inflammatory signals in the small intestinal mucosa of CD patients impacts epithelial GPX4 function. As suggested in point 3 of this reviewer, we tested the regulatory role of inflammatory stimuli on GPX4 expression and function. However, stimulation of IECs with a range of cytokines (IFN β , IFN γ , TNF α , IL-1 β , IL-22, IL-4, IL-6) did not impact GPX4 function (Supplementary Figure. 5F). We rather noted that intestinal epithelial GPX4 function is suppressed by dietary lipids (e.g. polyunsaturated fatty acids, Supplementary Figure 5A, B). We understand that this may not explain the differential regulation of epithelial GPX4 activity specifically in the inflamed small intestine of CD patients, however, based on the literature and our experiments, a distinct factor that would control GPX4 function in the inflamed small intestinal mucosa of CD patients remains elusive. As complex transcriptional differences between active CD and UC may occur in the mucosa, and specifically in intestinal epithelial cells and T-cells¹¹⁻¹⁴, we are currently unable to address this concern by compelling experimental evidence. We now discuss this in the revised manuscript.

Supplementary Figure 5. (F) Relative GPX4 enzymatic activity of MODE-K IECs stimulated with inflammatory cytokines for 24h (IL-1 β 10ng/ml, IL-4 10ng/ml, IL-6 20ng/ml, IL-22 10ng/ml, TNF α 50ng/ml, IFN β 2500U/ml and IFN γ 50ng/ml). One-way ANOVA Bonferroni’s multiple comparison test revealed no statistical differences.

“2. In figure 2 both omega-3 and omega-6 PUFA's elicit cytokine/chemokine responses from GPX4-deficient IEL. Shouldn't there be a difference inasmuch as omega-6 PUFAs are said to have harmful effects whereas omega-3 PUFAs are said not to have such effects. “

We agree that based on the current literature, omega-3 PUFAs are conceived anti-inflammatory while omega-6 PUFAs are thought to exert inflammatory actions^{15,16}. However, this concept was established in a GPX4 competent setting. In contrast, in the setting of *Gpx4*-deficiency, our data demonstrates that both, ω -3 and ω -6 PUFAs, but not monounsaturated fatty acids (i.e. oleic acid) or saturated fatty acids (i.e. palmitic acid) trigger lipid peroxidation and a cytokine response *only* in *Gpx4*-deficient IECs (Figure 2F-H). Scavenging lipid peroxidation with α -tocopherol or ferrostatin-1 ameliorated PUFA-induced cytokine production (Figure 4G, H), indicating a critical role for lipid peroxidation on the inflammatory phenotype. In line with this, α -tocopherol and liproxstatin-1 treatment ameliorated PUFA-induced enteritis (Figure. 5J, K). As such it appears that the availability (rather than the positioning) of double bonds within PUFAs define their propensity to fuel lipid peroxidation and an inflammatory response in GPX4-deficient IECs¹⁷. We now discuss this aspect in our revised manuscript.

“3. Does exposure of epithelial cells to pro-inflammatory cytokines such as TNF- α or IFN- γ have effects on GPX4 levels in such cells. “

We thank the reviewer for this comment which we partly discuss in point 1 of this reviewer's comment. Decreased GPX4 function specifically in IECs derived from the inflamed mucosa of CD patients suggests that the inflammatory milieu may account for the observed phenotype. However, stimulation of IECs with a range of cytokines did neither impact GPX4 expression nor function (Supplementary Figure. 5F and Figure A for this reviewer below). We rather noted that intestinal epithelial GPX4 function is suppressed by dietary lipids (e.g. polyunsaturated fatty acids, Supplementary Figure 5A, B). Importantly, we now also demonstrate that GPX4 did not control cytokine responses of IECs induced by TNF α or IL-1 β (Figure 2I, J). As such, it appears that GPX4 specifically restricted PUFA-induced cytokine responses while a range of cytokines did not affect GPX4 function.

Supplementary Figure 5. (F) Relative GPX4 enzymatic activity of MODE-K IECs stimulated with inflammatory cytokines for 24h (IL-1 β 10ng/ml, IL-4 10ng/ml, IL-6 20ng/ml, IL-22 10ng/ml, TNF α

50ng/ml, $\text{INF}\beta$ 2500U/ml and $\text{INF}\gamma$ 50ng/ml). One-way ANOVA Bonferroni's multiple comparison test revealed no statistical differences.

Figure A for reviewer: GPX4 expression and activity after stimulation with cytokines. (A) *Gpx4* expression of IECs stimulated with TNF α (50ng/ml), IL-1 β (10ng/ml) and INF γ (50ng/ml) for 24h by qPCR. N = 4. (B) Relative GPX4 enzymatic activity of MODE-K IECs stimulated with TNF α (50ng/ml), IL-1 β (10ng/ml) and INF γ (50ng/ml) for 24h. One-way ANOVA with Bonferroni correction revealed no significant differences.

Figure 2. (I, J) Quantification of IL-6 and CXCL1 expression from *siGpx4* and *siCtrl* IECs stimulated with TNF α or IL-1 β , for 24h by ELISA. N = 4. One-way ANOVA with Bonferroni correction revealed no significant differences.

“4. Mice with reduced GPX4 levels in epithelial cells have increased severity of DSS-colitis. This does not correspond to the fact that colonic cells in humans with have normal GPX4 levels and no inflammation of the colon was found in mice with reduced epithelial GPX4 levels on a PUFA-enriched diet. What accounts for this discrepancy?”

Indeed, *Gpx4*^{+/-IEC} mice exhibit increased DSS colitis severity compared to WT controls (Supplementary Figure 3J, K) while a PUFA-Western diet did not trigger colonic (but small intestinal) inflammation (Figure 3B-H). PUFAs are absorbed in the small intestine and not the colon of mice and humans¹⁸. As such one would expect that PUFAs elicit an inflammatory response in the small, but not the large intestine. In line with this, arachidonic acid and ferric iron exposure elicited small intestinal inflammation but no colonic inflammation (Figure 5D-F).

As this reviewer indicates, IECs from active UC patients exhibit GPX4 function that is comparable to healthy controls (Figure 1F, G) and epithelial GPX4 restricts colonic inflammation induced by DSS (Supplementary Figure 3I-K). The observation that GPX4 function is not differentially regulated in UC does not exclude a biological function of GPX4 in colonic inflammation. These two observations may not, for example, be linked because acute inflammation in a DSS model does not recapitulate every feature of chronic long-standing human UC. The DSS mouse model rather triggers acute inflammation by toxic damage of IECs consequent to a barrier breach. As such, DSS is a model of environmentally induced colitis and it appears plausible that *Gpx4*-deficient IECs are susceptible to cell death by cellular stressors. In line with this, we now demonstrate increased IEC death upon DSS exposure which could provide one explanation for worsened DSS-induced colitis in *Gpx4*^{+/-IEC} mice (Supplementary Fig. 3L, M). As such, the lack of GPX4 regulation in UC does not exclude a biological function that protects against DSS colitis in mice. We rather focused our study on PUFA-induced inflammatory processes (and did not follow up the DSS phenotype) as the mode of induction and the resulting inflammatory intestinal phenotype evoked by PUFAs appeared more relevant to us.

Supplementary Figure 3. *Gpx4*^{+/-IEC} mice are susceptible to DSS-induced colitis. (L) Representative images of TUNEL-labelled sections of indicated genotypes after DSS treatment (arrows denote brown TUNEL positive cells). N = 5-7. (M) Quantification of TUNEL⁺ IECs per crypt in indicated genotypes after DSS exposure. N = 5-7. Scale bar represents 100 μ m. * P < 0.05. unpaired two-tailed Students T-test was used.

“5. In Figure 3, neutrophils and other parameters of inflammation in WT mice not on a PUFA-enriched diet should be included to show if the diet elicits inflammation even in the absence of GPX4 deficiency.”

In order to address this, we isolated IEC scrapings from WT or *Gpx4*^{+/-IEC} mice on each respective diet and performed qPCR analysis for *Cxcl1*. Moreover, we stained MPO in our dietary groups. Indeed, neither a low-fat diet (LFD) nor a Western diet (WD) triggered *Cxcl1* expression or neutrophilic infiltration in WT or *Gpx4*^{+/-IEC} mice (Figure 3F and Supplementary Figure 4C). In contrast, a PUFA WD elicited *Cxcl1* expression and MPO⁺ cell infiltration in the mucosa in *Gpx4*^{+/-IEC} mice (Figure 3F and G). We now include these experiments in a revised manuscript.

Figure 3. PUFA enrichment of a Western diet induces focal enteritis with granuloma-like accumulation of inflammatory cells in *Gpx4*^{+/-IEC} mice. (F) Relative *Cxcl1* expression determined by qPCR. N = 8. * P < 0.05 (unpaired two-tailed Student’s *T* test) **(G)** Representative images of MPO⁺ cells in *Gpx4*^{+/-IEC} and WT mice on a PUFA-enriched WD. N = 5. The red arrows denote granuloma-like lesions with MPO⁺ cells (brown) intermingled between crypts marked with *. Scale bars indicate 50µm and 100µm. **(G)** *** P < 0.001 (one-way ANOVA with Bonferroni correction).

Supplementary Figure 4. (C) Representative immunohistochemistry images of MPO⁺ cells (brown) in indicated genotypes exposed to a chow diet, a low fat diet (LFD) and a Western diet (WD). N=4. There was no accumulation of MPO⁺ cells outside the vessels in the villus. Scale bar indicates 100μm.

“6. The detrimental effects of PUFA's in the GPX4-deficient mice is indeed contrary to what one might expect given the effect of PUFAs on the course of disease in humans, as the authors point out. It might be useful to test the effect of PUFAs in several strains of GPX4-deficient mice to see if there are strain differences in susceptibility to PUFA-induced injury in the face of GPX4 deficiency.”

We respectfully highlight, that we do not have other *Gpx4*-deficient mouse strains other than *Gpx4*^{+/-IEC} mice on a B6 background. As such, we are unable to perform the proposed experiment, although we agree that it would be informative.

Reviewer #2

“The submitted article,» Dietary lipids fuel GPX4-restricted enteritis resembling Crohn's disease» described the role of intestinal epithelial glutathione peroxidase 4 in controlling the gut homeostasis. Abstract: Expand the gene name ACSL4. In the key words, is acronym missing for the intestinal inflammation? “

We thank the reviewer for acknowledging the major theme of our work, namely the role of GPX4 in the control of gut homeostasis. We introduced the gene name of *Acs14* to the abstract. We do not use an acronym for intestinal inflammation.

“Introduction: The title is appropriate with a clear focus on dietary lipid-induced intestinal inflammation Crohn's disease. Authors have thoroughly reviewed the relevant literature. However, the importance of biolipid analysis and analytical techniques used in the detection and determination of lipids should be added. “

We thank the reviewer for his positive summary of our work and the notion that we have “thoroughly reviewed the relevant literature”. We now add information of biolipid analysis as detailed below.

“LC-MS/MS Methodology:

1. More information is needed for the extraction and quantification methods used for LC-MS/MS lipid analysis. A short method description in the supplemental data should be provided. “

The extraction protocol and analysis of bioactive lipids by LC-MS/MS were performed by the commercial provider Ambiotis SAS (Toulouse, France). Briefly cell pellets were treated with methanol and spiked with deuterated internal standards (ISTD) (PGE2-d4, 5-HETE-d8, LtB4-d4, LXA4-d5). After lysis and extraction at -20 °C for 1h the solution was centrifuged, supernatants were diluted with H₂O and subjected to solid phase extraction on an HLB 96 well plate (Oasis). Methyl formate-eluted lipid mediators were concentrated by a nitrogen stream evaporator prior to LC-MS/MS analysis. LC-MS/MS experiment was performed on a 1290 Infinity UHPLC system (Agilent Technologies, Santa Clara, CA, USA) mounted with a Kinetex Biphenyl column (2.1 mm, 50 mm, 1.7 μm, Phenomenex), maintained at 50°C. The UHPLC system was coupled to a 6490 triple quadrupole MS (Agilent Technologies), equipped with electrospray ionization source. Analyses were performed in negative ion mode with in multiple reaction monitoring detection mode by use of nitrogen as collision gas. Peak detection, integration, and quantitative analysis were done by use of MassHunter Quantitative Analysis Software (Agilent Technologies). Results are expressed in a quantitative manner as pg/million cells. Validation and extraction efficacy of the extraction protocol was validated in Le Faouder et al. ¹⁹.

This is now included in the methods section, page 26.

“2. The authors did not state clearly the results were semi or absolute quantitation of lipids. As the two references that were provided showed that the method is semi-quantitative analysis of lipids, while Ambiotis

SAS (Toulouse, France) lipidomics core facility claims absolute quantification of lipids on their webpage. “

For quantification, standard curves were generated with authentic standards (a known amount). The results obtained for lipid mediator quantification are expressed in a quantitative manner (i.e. pg/million cells), which we now indicated in the y-axis of each figure. For conducting the absolute quantification ISTD responses were set in relation to their non-labelled sample-derived counterparts. We now also specifically state that the results are absolute quantitation of lipids in the methods section.

“3. According to the reference 59, the method is fully validated with necessary experiments. There is no clarity that the authors have reproduced all the experiments and results or some part. A brief description about the validation experiments is needed including the results. “

As reported in Le Faouder et al¹⁹, the preparation of samples was validated with the extraction efficiency and the matrix effect. Briefly, the samples (n=3 for each experimental group) were spiked with ISTD and then processed as described in point 1. In parallel, a separate set of STD/ISTD solution prepared in MeOH was directly analysed in triplicate. The extraction efficiency was determined as the percent difference between pre-spiked and post-spiked samples. Sample treatment was validated using the recovery efficacy of each standard diluted in cell medium at two concentrations (250 pg and 62.5 ng/mL) and analyzed in triplicate. The CVs and recovery values for this method were published previously¹⁹. QC samples (n=5) were spiked with STD at two concentrations (100 and 20 ng/mL) and ISTD. These QC samples were then processed as a biological sample described in point 1. Sample treatment was thus validated using the CV and the accuracy obtained for each standard. The cv and accuracy values are listed below (for this reviewer):

Table for reviewer. %CV and %ACC obtained for extraction validation. One representative molecule of each family shown.

	QCM		QCH	
	%CV (n=5)	%ACC	%CV (n=5)	%ACC
6-K-PGF1A	0,60	107,37	7,86	97,53
TXB2	2,47	110,09	0,04	111,21
PGE2	7,13	108,77	0,43	112,38
LXA4	4,00	107,15	5,95	104,06
LTB4	6,04	100,16	3,28	106,21
12-HETE	7,39	102,00	8,78	109,24

The %CV are clearly below 15 % and the %ACC between 85 and 115 %. We conclude that results obtained with this method are reproducible and close to the expected values. The results of our analysis are shown in Figure 6A-C.

“4. The authors need to clarify on the purpose of using Ambiotis SAS (Toulouse, France) standard operating procedure in the analyses. Provide details on the parameters or part used from this procedure. “

Ambiotis used the work by Le Faouder et al¹⁹ and generated a standard operating procedure to provide a reproducible analytical platform that is commercially available. From our perspective, we have had

good experiences with this service, communication with the company and the transparency of the protocols that have been carried out. The method used by Ambiotis is now provided in the methods section as detailed in response to point 1 of this reviewer.

“5. The authors have not clarified on the type of samples, number of biological replicates used per group, names and concentrations of complete list of analytes (lipids). Provide the list as a supplementary table.”

We now provide these information in supplementary table 3 below.

Supplementary Table 3. Lipid mediators [pg/million cells] determined by LC-MS/MS. Samples were obtained from *siCtrl* and *siGpx4* WT with or without co-deletion of *Acsl4*^{-/-} IECs and analysed following stimulation with AA or vehicle for 24h. Three independent experiments were performed and pooled for the analysis.

			COX		15-LOX/5-LOX		5-LOX		15-LOX		12-LOX
			TXB2	6-K-PGF1A	PGE2	LXB4	LXA4	LTB4	5-HETE	15-HETE	12-HETE
WT	vehicle	siCtrl	42,04	16,15	1081,85	39,19	0,70	-	21,83	326,15	39,33
		siGpx4	59,27	17,77	1956,48	39,32	0,00	0,13	21,96	569,22	39,40
	AA	siCtrl	187,07	105,14	8283,86	42,15	5,51	-	583,06	3663,83	568,19
		siGpx4	241,29	129,45	9992,21	13,13	2,05	-	455,87	4338,66	472,38
Acsl4 ^{-/-}	vehicle	siCtrl	112,76	84,33	4519,04	64,05	0,54	0,26	47,10	1039,80	57,85
		siGpx4	123,74	103,39	5955,85	49,22	0,87	-	67,26	1527,85	80,87
	AA	siCtrl	498,36	426,47	27475,92	135,27	7,44	0,76	3670,79	9504,63	1072,46
		siGpx4	431,44	398,61	27953,24	118,42	11,01	-	2645,51	8468,00	686,80

			CYP450							
			14,15-DHET	11,12-DHET	8,9-DHET	5,6-DHET	14,15-EET	11,12-EET	8,9-EET	5,6-EET
WT	vehicle	siCtrl	1,11	4,03	-	-	11,05	12,82	99,73	84,57
		siGpx4	1,50	5,01	1,57	-	38,88	10,54	145,25	79,39
	AA	siCtrl	1404,18	878,23	400,89	24,89	1801,47	445,98	671,40	394,19
		siGpx4	812,48	754,14	240,99	16,21	1850,24	558,16	518,50	218,72
Acsl4 ^{-/-}	vehicle	siCtrl	2,06	3,51	-	-	24,60	-	228,78	233,70
		siGpx4	2,12	10,07	-	-	9,08	12,15	251,82	94,74
	AA	siCtrl	2478,41	3133,22	1374,39	96,98	6648,23	2349,70	1977,08	1655,59
		siGpx4	1829,65	2542,62	913,63	70,73	5630,30	1830,04	1838,77	1401,31

“6. LC-MS/MS raw data should be deposited in public repositories e.g., MetaboLights etc. “

We are unable to deposit the data in MetaboLights as the company refused to do so. Ambiotis reasoned that this would offend their “confidentiality agreements as a commercial provider”. Instead, we uploaded the data in a publicly available platform (<https://data.mendeley.com>) which we also included in our methods section (DOI: 10.17632/k9ync2kd3g.2; Data for: Dietary lipids fuel GPX4-restricted enteritis resembling Crohn’s disease).

Reviewer #3

“In this work, the authors studied how lipid peroxidation which is efficiently controlled by glutathione peroxidase 4 (GPX4) may impact on Crohn's disease (CD), a form of inflammatory bowel disease. This study is of particular relevance given the reported association of a single nucleotide polymorphism within GPX4 and CD. The authors now show that intestinal epithelial-specific ablation of Gpx4 in mice subjected to PUFA-enriched Western diet develop some form of neutrophilic enteritis which can be compensated by dietary vitamin E supplementation. Importantly, the authors also provide evidence that CD patient biopsies display reduced GPX4 activity and markers of lipid peroxidation. Overall, this is a very interesting and well-performed study providing mechanistic insights into the role of GPX4 in the development of CD. “

We thank the reviewer for highlighting the “particular relevance” and that we provide data on *Gpx4*-deficiency in human CD. We are grateful for his/her interest in our study and the very positive feedback.

“- Page 6: The authors did not obtain intestinal epithelia-specific homozygous Gpx4 null mice as these mice seem to die during embryogenesis - did the authors consider to rescue embryonic lethality by supplementing the diet with high vitamin E. What is the actual vitamin E content of the diet used in light of various papers showing that some KO models for GPX4 can be actually rescued by high vitamin E concentration. It may worth trying but at least it should be discussed here. “

Indeed, our mouse chow diet contains 150mg α -tocopherol per kg diet which approximates a daily intake of ~22.5mg/kg/mouse (assuming that a 20g mouse consumes 3g chow diet a day) which is approximately ~10-100x more than what is found in a human diet (with regional differences, see WHO report „Vitamin and mineral requirements in human nutrition“, Second edition). This α -tocopherol concentration in chow diet is apparently insufficient to allow generation of viable homozygous *Gpx4*^{-/-IEC} mice. In collaboration with Prof. Christoph Becker we report that supplementing the diet of mothers during gestation with α -tocopherol (500 mg/kg Tocopheryl Acetate) led to a partial rescue of embryonic lethality. Specifically, we noted that 11% of *Gpx4*^{-/-IEC} offspring (i.e. *Gpx4*^{fl/fl}; *Villin Cre*^{+/-}) (Supplementary Figure 2A, B) was born when the mother was exposed to α -tocopherol (instead of expected 25%) (Supplementary Table 1). Of note, *Gpx4*^{-/-IEC} offspring had a significantly lower weight at birth compared to WT littermates but regained weight during the following weeks (Supplementary Figure 2C-F). Histopathological analyses of adult GPX4 knockout and control mice showed no spontaneous intestinal phenotype (Supplemental Figure 2G, H). We now included these results in the revised manuscript.

Supplementary Table 1: α -tocopherol rescues embryonic lethality of $Gpx4^{-IEC}$ mice. Expected (Mendelian ratios) and observed genotypes after cross breeding $Gpx4^{fl/wt} Villin Cre^+$ and $Gpx4^{fl/fl} Villin Cre^-$ parents fed with α -tocopherol enriched diet. Overall, 74 newborn animals were analyzed.

$Gpx4^{fl/wt} Villin Cre^+$ X $Gpx4^{fl/fl} Villin Cre^-$	$Gpx4^{fl/wt}$ $Villin Cre^-$	$Gpx4^{fl/wt}$ $Villin Cre^+$	$Gpx4^{fl/fl}$ $Villin Cre^-$	$Gpx4^{fl/fl}$ $Villin Cre^+$
Expected	25%	25%	25%	25%
Observed	30%	32%	27%	11%

Supplementary Figure 2

Supplementary Figure 2. α -tocopherol supplementation rescues embryonic lethality of *Gpx4*^{-/-IEC} mice. (A, B) Representative GPX4 immunoblot from isolated IECs and (B) representative confocal microscopy images of GPX4 labelled small intestinal sections of indicated genotypes on an α -tocopherol supplemented diet. Actin served as loading control in (A). N = 3. (C, D) Representative images of WT and *Gpx4*^{-/-IEC} mice born to mothers exposed to an α -tocopherol enriched diet until weaning. N = 3. (E) Weight course of *Gpx4*^{-/-IEC} and WT mice on an α -tocopherol enriched diet. N = 3. Note that *Gpx4*^{-/-IEC} mice show a reduced body weight one week after birth which they regained after 7 weeks. (F) Body weight at 1 week of age of WT and *Gpx4*^{-/-IEC} mice born to a mother that was exposed to an α -tocopherol enriched diet. Each dot represents an individual animal. (G, H) Representative H&E images of indicated genotypes from the small (G) and large intestine (H). Scale bars indicate 100 μ m. N = 3. * P < 0.05 *** P < 0.001. Unpaired two-tailed Students *T*-test.

“- Figure 3: Did the authors try to rescue diet-induced focal enteritis with cognate ferroptosis inhibitors such as ferrostatin-1 or liproxstatin-1 similar to what was done with alpha-tocopherol supplementation in fig. 5? Would leukocyte/neutrophil infiltration be sensitive to such a treatment? “

We now perform a 3-month PUFA Western diet with liproxstatin-1 treatment from week 6 onwards. Similar to α -tocopherol treatment (previously included in the manuscript Figure 5J), liproxstatin-1 treatment ameliorated PUFA-induced enteritis in *Gpx4*^{+/-IEC} mice as assessed by H&E analysis (Figure 5K). Similarly, liproxstatin-1 protected against MPO⁺ neutrophil infiltration in *Gpx4*^{+/-IEC} mice (Supplementary Figure 10C). We now included these results in the revised manuscript.

Figure 5. (K) Enteritis histology score of WT and *Gpx4*^{+/-IEC} mice exposed to a PUFA-enriched WD (PUFA WD) for 3 months with and without liproxstatin-1 treatment intraperitoneally from 6 weeks onwards [10mg/kg] until the closure of the experiment. Each dot represents one experimental animal. Median shown. *** P < 0.001. one-way ANOVA with Bonferroni’s multiple comparison test was used.

Supplementary Figure 10. (C) Representative images of MPO⁺ cells (brown) in *Gpx4*^{+/-IEC} and WT mice on a PUFA-enriched WD with or without liproxstatin-1 treatment. N = 5. Scale bars indicate 50µm.

“- Figure 4: the authors try to make a link between lipoxygenase activity and ferric iron induced cytokine responses. As most «lipoxygenase-specific» inhibitors display non-specific radical trapping activity (RTA) (!) one may wonder whether there is an actual contribution of lipoxygenase(s) to the findings presented here. As long as the authors do not rule out that these compounds harbor RTA activity (see Shah et al ACS Cent Sci 2018), it cannot be stated here (see also page 12, last paragraph). Otherwise data on knockdown/knockout of one or several lipoxygenase(s) should be included here. “

We appreciate this concern and now performed co-silencing experiments for *Alox12* and *Alox15* in AA-stimulated *Gpx4*-deficient IECs. Co-silencing of *Alox12* and *Alox15* (Supplementary Figure 7I, J) ameliorated AA-induced CXCL1 expression in *Gpx4*-deficient IECs (Figure 4K). We thank the reviewer for raising this important point and we now include these results in the revised manuscript.

Figure 4. (K) Quantification of CXCL1 in the supernatant from *siGpx4* and *siAlox12* or *siAlox15* co-silenced IECs stimulated with AA (20µM) for 24h. N = 4. ** P < 0.01, (one-way ANOVA with Bonferroni's multiple comparison test).

Supplementary Figure 7. (I) Relative *Alox12* expression of siRNA silenced MODE-K IECs determined by qPCR. N = 4. **(J)** Relative *Alox15* expression of siRNA silenced MODE-K IECs determined by qPCR. N = 4. * P < 0.05, ** P < 0.01. Unpaired two-tailed Student's *T*-test

“- *Figure 5: the authors may indicate how much alpha-tocopherol was supplemented in the drinking water? Would ferroptosis inhibitors confer similar effects (see my comment in the foregoing)? - For the general reader, it might be appropriate to include a model illustrating the main effects described here.*”

We indicated that we supplemented 0.4mg/ml α -tocopherol in drinking water. We addressed that liproxstatin-1 ameliorates enteritis in *Gpx4*^{+/-IEC} mice on a PUFA WD (see point above). We have included a mode of GPX4-restricted cytokine restriction in Supplementary Figure 12.

Supplementary Figure 12. Model of PUFA-induced and GPX4-restricted enteritis. Left panel: Exposure of polyunsaturated fatty acids (PUFAs) such as arachidonic acid (AA) induces LOX-mediated lipid peroxidation (LPO) that is fuelled by iron availability. GPX4 critically restricts epithelial LPO to maintain gut homeostasis ²⁰. **Right panel:** Reduced intestinal epithelial GPX4 function is a feature of ileal Crohn's disease and may be evoked by genetic or environmental cues. In IECs with reduced GPX4 function, PUFAs such as AA promote lipoxygenase-driven LPO and cytokine production (i.e. CXCL1 and IL-6 expression), which are fuelled by iron availability and ameliorated by the LPO scavenger α -tocopherol. ACSL4 may limit a PUFA-induced cytokine response in IECs with reduced GPX4 activity by modulation of AA metabolism. As such, similar mechanisms that control ferroptosis ²⁰ also control epithelial cytokine responses of IECs with reduced GPX4 function. A PUFA Western diet triggers granuloma-like neutrophilic intestinal inflammation resembling human CD which can be ameliorated by α -tocopherol treatment. As such, GPX4-restricted LPO emerges as a rheostat of PUFA-induced neutrophilic inflammation in the intestine which is fuelled by dietary lipids.

“ Minor: - Page 3, first paragraph: the first reports showing that GPX4 protects from lipid peroxidation and non-apoptotic cell death are Ursini et al. 1982 and Seiler et al. Cell Metab 2008, respectively. - Page 5, it should read Figure 2F and Figure 2G,H. “

We thank the reviewer for highlighting these references. We have edited the references accordingly, and we have corrected the figure labelling.

References

1. Lev-Tzion, R., Griffiths, A. M., Leder, O. & Turner, D. Omega 3 fatty acids (fish oil) for maintenance of remission in Crohn's disease. *The Cochrane database of systematic reviews*, CD006320 (2014).
2. Belluzzi, A. *et al.* Effect of an enteric-coated fish-oil preparation on relapses in Crohn's disease. *N. Engl. J. Med.* **334**, 1557-1560 (1996).
3. Feagan, B. G. *et al.* Omega-3 free fatty acids for the maintenance of remission in Crohn disease: the EPIC Randomized Controlled Trials. *JAMA* **299**, 1690-1697 (2008).
4. Manson, J. E. *et al.* Marine n-3 Fatty Acids and Prevention of Cardiovascular Disease and Cancer. *N. Engl. J. Med.* **380**, 23-32 (2019).
5. Hull, M. A. *et al.* Eicosapentaenoic acid and aspirin, alone and in combination, for the prevention of colorectal adenomas (seAFOod Polyp Prevention trial): a multicentre, randomised, double-blind, placebo-controlled, 2 x 2 factorial trial. *Lancet* **392**, 2583-2594 (2018).
6. Limketkai, B. N. *et al.* Dietary interventions for induction and maintenance of remission in inflammatory bowel disease. *The Cochrane database of systematic reviews* **2**, CD012839 (2019).
7. Gorard, D. A. *et al.* Initial response and subsequent course of Crohn's disease treated with elemental diet or prednisolone. *Gut* **34**, 1198-1202 (1993).
8. Yamamoto, T., Nakahigashi, M., Umegae, S., Kitagawa, T. & Matsumoto, K. Impact of elemental diet on mucosal inflammation in patients with active Crohn's disease: cytokine production and endoscopic and histological findings. *Inflamm. Bowel Dis.* **11**, 580-588 (2005).
9. Penagini, F. *et al.* Nutrition in Pediatric Inflammatory Bowel Disease: From Etiology to Treatment. A Systematic Review. *Nutrients* **8** (2016).
10. Gunasekera, V., Mendall, M. A., Chan, D. & Kumar, D. Treatment of Crohn's Disease with an IgG4-Guided Exclusion Diet: A Randomized Controlled Trial. *Dig. Dis. Sci.* **61**, 1148-1157 (2016).
11. Howell, K. J. *et al.* DNA Methylation and Transcription Patterns in Intestinal Epithelial Cells From Pediatric Patients With Inflammatory Bowel Diseases Differentiate Disease Subtypes and Associate With Outcome. *Gastroenterology* **154**, 585-598 (2018).
12. Fang, K., Grisham, M. B. & Keivil, C. G. Application of Comparative Transcriptional Genomics to Identify Molecular Targets for Pediatric IBD. *Front. Immunol.* **6**, 165 (2015).
13. Lee, J. C. *et al.* Gene expression profiling of CD8+ T cells predicts prognosis in patients with Crohn disease and ulcerative colitis. *J. Clin. Invest.* **121**, 4170-4179 (2011).
14. Haberman, Y. *et al.* Ulcerative colitis mucosal transcriptomes reveal mitochondriopathy and personalized mechanisms underlying disease severity and treatment response. *Nature communications* **10**, 38 (2019).
15. Ganesan, B., Brothersen, C. & McMahon, D. J. Fortification of foods with omega-3 polyunsaturated fatty acids. *Crit. Rev. Food Sci. Nutr.* **54**, 98-114 (2014).
16. Ungaro, F., Rubbino, F., Danese, S. & D'Alessio, S. Actors and Factors in the Resolution of Intestinal Inflammation: Lipid Mediators As a New Approach to Therapy in Inflammatory Bowel Diseases. *Front. Immunol.* **8**, 1331 (2017).
17. Ayala, A., Munoz, M. F. & Arguelles, S. Lipid peroxidation: production, metabolism, and signaling mechanisms of malondialdehyde and 4-hydroxy-2-nonenal. *Oxid. Med. Cell. Longev.* **2014**, 360438 (2014).
18. Yang, Q. *et al.* Dietary intake of n-3 PUFAs modifies the absorption, distribution and bioavailability of fatty acids in the mouse gastrointestinal tract. *Lipids Health Dis.* **16**, 10 (2017).
19. Le Faouder, P. *et al.* LC-MS/MS method for rapid and concomitant quantification of pro-inflammatory and pro-resolving polyunsaturated fatty acid metabolites. *J. Chromatogr. B Analyt. Technol. Biomed. Life Sci.* **932**, 123-133 (2013).

20. Stockwell, B. R. *et al.* Ferroptosis: A Regulated Cell Death Nexus Linking Metabolism, Redox Biology, and Disease. *Cell* **171**, 273-285 (2017).

Reviewers' Comments:

Reviewer #1:

Remarks to the Author:

In this re-submission of their manuscript the authors have included several data that address the initial concerns. Nevertheless, certain concerns remain and require some further changes.

1. The authors refer to the histopathologic changes observed on Gpx4-partially deficient mice as "Crohn's-like" in character. However, it appears that the main inflammatory change is an influx of neutrophils which is only one feature of Crohn's disease. The authors need to provide data that the inflammation is also characterized by an increase in dendritic cell/macrophage infiltration and that the latter is associated with the presence increased amounts of pro-inflammatory cytokines such as IL-12, IL-23, IFN-g and IL-17.

2. If one compares the amount of IL-6 elicited by epithelial cell exposure to AA with that elicited by TNF α , one notes that the latter elicits far more inflammatory cytokine (Figure 2). This implies that while AA (and other PUFAs) have definite effects in the absence of Gpx4, the latter are quantitative unimportant with respect to the induction of inflammation occurring in small intestinal Crohn's disease. In other words, inflammation would be present in the absence of a PUFA effect and, more importantly, limitation of PUFA from the diet would have a minor effect on the overall inflammation. Ideally, the authors would address this issue experimentally by studies of mice lacking cytokine function that might provide insight into the effect of epithelial cell cytokine secretion alone. Alternatively, they need to discuss the significance of their findings in terms of the magnitude of the effect.

3. Related to the comment above, if the concern that the Gpx4 deficiency is limited to the small intestine in humans with Crohn's disease and is not seen in ulcerative colitis, a large intestinal disease. This observation clearly shows that such deficiency is clearly not a primary defect in IBD and, at best, is an aggravating factor in small intestinal disease. As this may ultimately limit the efficacy of IBD treatment by reducing PUFA intake or α -tocopherol treatment and deserves full Discussion.

Reviewer #2:

Remarks to the Author:

Thank you for the revised manuscript. The authors have addressed my points satisfactorily.

Reviewer #3:

Remarks to the Author:

The authors have done a great job and addressed all my concerns and have included a substantial amount of new in vivo data. It is indeed remarkable that embryonic lethality of endothelial specific Gpx4 null embryos can be rescued by vitamin E supplementation.

I just have a minor comment as it is not always clear whether the authors refer to GPX4 expression, activity and/or function? This should be unequivocally stated in the text and in the figure legends.

Beyond this, I have no further comments.

Reviewer #1:

In this re-submission of their manuscript the authors have included several data that address the initial concerns. Nevertheless, certain concerns remain and require some further changes.

We thank the reviewer for thoroughly reviewing our manuscript and for allowing us to address his/her concerns as detailed below.

1. The authors refer to the histopathologic changes observed in *Gpx4*-partially deficient mice as “Crohn’s-like” in character. However, it appears that the main inflammatory change is an influx of neutrophils which is only one feature of Crohn’s disease. The authors need to provide data that the inflammation is also characterized by an increase in dendritic cell/macrophage infiltration and that the latter is associated with the presence of increased amounts of pro-inflammatory cytokines such as IL-12, IL-23, IFN- γ and IL-17.

We agree with the reviewer that mucosal neutrophilic inflammation in Crohn’s disease is one (arguably early) feature of this disease¹ and that there is a plethora of other cellular and molecular characteristic events that underlie the pathogenesis of this complex heterogeneous disorder. We reasoned that neutrophilic inflammation in mice that lack one *Gpx4* allele resembles human Crohn’s disease as:

- (I) intestinal epithelial cells from Crohn’s disease patients exhibit impaired GPX4 activity which we modelled in mice.
- (II) *Gpx4*^{+/-IEC} mice displayed several unique histopathological Crohn’s disease features (i.e. neutrophil accumulation in granuloma-like structures, submucosal infiltration of inflammatory cells, crypt hyperplasia and epithelial injury (Figure 3B, C)).
- (III) histopathologic features were accompanied by mucosal *IL-8* homologue expression (Figure 3F), as similarly reported for the mucosa of Crohn’s disease patients¹.

We agree with this reviewer that these features do not entirely “recapitulate” human CD and that the inflammatory response in PUFA Western diet exposed *Gpx4*^{+/-IEC} mice is likely more complex. We appreciate that our report does not comprise a complete characterisation of the immunophenotypic landscape which would support the notion that this model entirely “recapitulates” human Crohn’s disease. However, we are convinced that the reported phenotype “resembles” human Crohn’s disease (see above), as this wording implies that some (but not necessarily all) Crohn’s disease features can be found in the reported model.

We have performed F4/80 immunohistochemical labelling of macrophages in the mucosa of PUFA Western diet fed *Gpx4*^{+/-IEC} mice (Figure for reviewer below). As we are currently unable to provide a large-scale immune-phenotype (i.e. flow cytometry of immune cell populations and cytokine profiles), the significance of macrophage infiltration is unclear to us. Therefore, we believe that the addition of these data is descriptive and does not help to improve our manuscript as it stands now. However, we agree that a deep understanding of the molecular mechanism in this model would be highly informative. We respectfully suggest that this approach (immunophenotyping by flow cytometry and combined cytokine profiling) is beyond the scope of the present manuscript as this requires generation, experimentation and analysis of mice that would take >8 months, and as this experiment has *not* been requested by any reviewer in the first revision process. However, we now temper our wording in the revised manuscript and discuss that future studies are needed to explore the more

complex immune cell profile which may set the basis for a detailed immunophenotypic landscape in this disease model.

Figure for reviewer: (A) Representative images of F4/80⁺ macrophage infiltration in the small intestinal mucosa of PUFA Western diet fed WT and *Gpx4*^{+/-IEC} mice. N=5 Scale bar indicates 50µm.

2. If one compares the amount of IL-6 elicited by epithelial cell exposure to AA with that elicited by TNF α , one notes that the latter elicits far more inflammatory cytokine (Figure 2). This implies that while AA (and other PUFAs) have definite effects in the absence of *Gpx4*, the latter are quantitative unimportant with respect to the induction of inflammation occurring in small intestinal Crohn's disease.

We agree to the careful observation that the absolute IL-6 concentration in the supernatant of TNF-stimulated intestinal epithelial cells (IECs) with reduced GPX4 activity (*siGpx4*) is ~20 times higher than the IL-6 concentration after PUFA stimulation (Figure 2G and Figure 2I). In contrast, TNF-induced CXCL1 production is comparable to PUFA-induced CXCL1 production of *siGpx4* IECs (Figure 2H and Figure 2J). While these observations "imply" that CXCL1 may be more relevant in our inflammatory model, we would argue that the relevance of both cytokines are yet undetermined (in our model and also in human Crohn's disease). As it stands now, we respectfully suggest that the biological relevance of these cytokines in intestinal inflammation in mice or human Crohn's disease cannot be deduced from the level of expression in our cell model, because the impact of these cytokines on the *in vivo* phenotype (in mice or humans) is undetermined. We now state this in the discussion. Notably, IL-6 appears to drive some aspects of Crohn's disease, as anti-IL-6 antibody treatment may induce clinical response and remission in treatment-refractory patients².

In other words, inflammation would be present in the absence of a PUFA effect and, more importantly, limitation of PUFA from the diet would have a minor effect on the overall inflammation. Ideally, the authors would address this issue experimentally by studies of mice lacking cytokine function that might provide insight into the effect of epithelial cell cytokine secretion alone. Alternatively, they need to discuss the significance of their findings in terms of the magnitude of the effect.

We do not observe intestinal inflammation in our model in the absence of PUFA supplementation (i.e. after feeding a standard Western diet). Therefore, we studied PUFAs as *trigger* for GPX4-restricted intestinal inflammation in cells and mice. Whether reduction of PUFAs from the human diet has beneficial effects for mice or patients with *established* inflammation (and probably a more complex inflammatory milieu) requires future clinical (and equivalent preclinical) studies. We agree also with

this reviewer that identification of critical cytokine drivers (by genetic studies) is of importance for understanding the reported phenotype, however, we respectfully suggest that this is beyond the scope of the current manuscript. We now discuss this and the magnitude of effect of measured cytokines as suggested by this reviewer.

3. Related to the comment above, if the concern that the Gpx4 deficiency is limited to the small intestine in humans with Crohn's disease and is not seen in ulcerative colitis, a large intestinal disease. This observation clearly shows that such deficiency is clearly not a primary defect in IBD and, at best, is an aggravating factor in small intestinal disease. As this may ultimately limit the efficacy of IBD treatment by reducing PUFA intake or α -tocopherol treatment and deserves full Discussion.

We agree with the reviewer that reduced GPX4 function is observed in intestinal epithelial cells of the lesional small intestine in Crohn's disease, but not in ulcerative colitis. We also agree with his/her notion that (I) we never claimed that reduced GPX4 function arises from primary defects (e.g. genetic variation within *GPX4*) and (II) that yet undetermined cellular hubs in inflamed IECs rather control GPX4 function. For these reasons, we initially chose a model with reduced, but not completely abrogated GPX4 function (i.e. deletion of one *Gpx4* allele). We now discuss this more precisely.

Reviewer #2:

Thank you for the revised manuscript. The authors have addressed my points satisfactorily.

We thank the reviewer for his/her input and interest in our study.

Reviewer #3:

The authors have done a great job and addressed all my concerns and have included a substantial amount of new in vivo data. It is indeed remarkable that embryonic lethality of endothelial specific Gpx4 null embryos can be rescued by vitamin E supplementation.

We thank the reviewer for his/her input and interest in our study.

I just have a minor comment as it is not always clear whether the authors refer to GPX4 expression, activity and/or function? This should be unequivocally stated in the text and in the figure legends.

We thank the reviewer for highlighting this issue which we corrected throughout the manuscript.

References

1. Fournier, B. M. & Parkos, C. A. The role of neutrophils during intestinal inflammation. *Mucosal Immunol.* **5**, 354-366 (2012).
2. Danese, S. *et al.* Randomised trial and open-label extension study of an anti-interleukin-6 antibody in Crohn's disease (ANDANTE I and II). *Gut* **68**, 40-48 (2019).

Reviewers' Comments:

Reviewer #1:

Remarks to the Author:

The authors in this review round did not fully respond to my critique. They show that PUVA exposure in Gpx-4-partially deficient mice results in a Crohn's-like small intestinal histologic picture. This is associated with increased CXCL1 production, a neutrophil specific chemokine. IL-6 induction by TNF- α was relatively weak.

The data therefore suggest that PUVA exposure can cause some aspects of full-blown Crohn's disease. As such, it contributes to the understanding of some aspects of Crohn's disease.

Point-to-point response NCOMMS-19-16671

Reviewer #1

The authors in this review round did not fully respond to my critique. They show that PUVA exposure in Gpx-4-partially deficient mice results in a Crohn's-like small intestinal histologic picture. This is associated with increased CXCL1 production, a neutrophil specific chemokine. Il-6 induction by TNF- α was relatively weak.

The data therefore suggest that PUVA exposure can cause some aspects of full-blown Crohn's disease. As such, it contributes to the understanding of some aspects of Crohn's disease.

We thank the reviewer for summarising our work and for highlighting the novelty. We have edited the manuscript accordingly, to avoid overinterpretation of our study.